# Improving the usability of the Multi-angle Imaging SpectroRadiometer (MISR) L1B2 Georectified Radiance Product (2000–present) in land surface applications

Michel M. Verstraete[1], Linda A. Hunt[2], and Veljko M. Jovanovic[3]

[1]Global Change Institute (GCI), University of the Witwatersrand, Braamfontein, Republic of South Africa.
[2]Science Systems and Applications, Inc. (SSAI), Hampton, VA 23666-5845, USA.
[3]NASA Jet Propulsion Laboratory (JPL), Pasadena, CA 91109, USA.

**Correspondence:** Michel M. Verstraete (Michel.Verstraete@wits.ac.za or MMVerstraete@gmail.com)

**Abstract.** The Multi-angle Imaging SpectroRadiometer (MISR) instrument on NASA's Terra platform has been acquiring global measurements of the spectro-directional reflectance of the Earth since 24 February 2000 and is still operational as of this writing. The primary radiometric data product generated by this instrument is known as the Level 1B2 Georectified Radiance Product (GRP): it contains the 36 radiometric measurements acquired by the instrument's nine cameras, each observing the planet in four spectral bands. The product version described here is projected on a digital elevation model and is available from the NASA Langley Atmospheric Science Data Center (ASDC) (https://doi.org/10.5067/Terra/MISR/MI1B2T_L1.003; Jovanovic et al. (1999)). The MISR instrument is highly reliable. Nevertheless, its on-board computer occasionally becomes overwhelmed by the amount of raw observations coming from the cameras' focal planes, especially when switching into or out of LOCAL MODE acquisitions that are often requested in conjunction with field campaigns. Whenever this occurs, one or more lines of data are dropped while the computer resets and readies itself for accepting new data. Although this type of data loss is minuscule compared to the total amount of measurements acquired, and is marginal for atmospheric studies dealing with large areas and long periods of time, this outcome can be crippling for land surface studies that focus on the detailed analysis of particular scenes at specific times. This paper describes the problem, reports on the prevalence of missing data, proposes a practical solution to optimally estimate the values of the missing data and provides evidence of the performance of the algorithm through specific examples in Southern Africa. The software to process MISR L1B2 GRP data products as described here is openly available to the community from the GitHub web site https://github.com/mmverstraete or https://doi.org/10.5281/zenodo.3519988. Two additional sets of resources are also made available on the research data repository of GFZ Data Services in conjunction with this paper. The first set (A; Verstraete et al., 2020; https://doi.org/10.5880/fidgeo.2020.012) includes five items: (A1), a compressed archive, `L1B2_Out.zip`, containing all intermediary, final and ancillary outputs created while generating the figures of this paper; (A2), a user manual, `L1B2_Out.pdf`, describing how to install, uncompress and explore those files; (A3), an additional compressed archive, `L1B2_Suppl.zip`, containing a similar set of results, only for eight other sites, spanning a much wider range of geographical, climatic and ecological conditions; (A4), a companion user manual, `L1B2_Suppl.pdf`, describing how to install, uncompress and explore those additional files; and (A5), a separate input MISR data archive, `L1B2_input_68050.zip`, for PATH 168, ORBIT 68050. This latter archive is

usable with (B), the second set (Verstraete and Vogt, 2020; https://doi.org/10.5880/fidgeo.2020.011), which includes (B1), a stand-alone, self-contained, executable version of the L1B2 correction codes, `L1B2_Soft_Win.zip`, using the IDL Virtual Machine technology that does not require a paid IDL license, as well as (B2), a user manual, `L1B2_Soft_Win.pdf`, to explain how to install, uncompress and use this software.

*Copyright statement.* TEXT

# 1 Introduction

The Multi-angle Imaging SpectroRadiometer (MISR) is one of the five Earth observing instruments hosted on the NASA Terra platform, launched into a sun-synchronous low-Earth orbit on 18 December 1999 (altitude: 713 km; equatorial crossing time: 10:30 AM, local time). The orbit of Terra is precisely maintained so that, after completing 233 revolutions, the satellite observes again the same locations on Earth from the same directions: this cycle takes 16 days, so the satellite performs 14.6 orbits per day. Each of those systematically repeating tracks is called a PATH. MISR features nine cameras pointing at various bore angles, both forward, nadir and aft of the platform (DF: 70.3°, CF: 60.2°, BF: 45.7°, AF: 26.2°, AN: 0.1°, AA: 26.2°, BA: 45.7°, CA: 60.2°, and DA: 70.6°, each one equipped to acquire data in four spectral bands of 20–40 nm full width at half height: blue (446.4 nm), green (557.5 nm), red (671.7 nm), and near-infrared (866.4 nm)), for a total of 36 data channels. MISR became operational on 24 February 2000 and is still collecting data as of this writing. This instrument has been extensively described in the literature (Diner et al. (1989); Diner et al. (1998); Diner et al. (1999a); Diner et al. (2002); Diner et al. (2005)), and more recently in a special issue of the MDPI journal *Remote Sensing* dedicated to that instrument (See https://www.mdpi.com/journal/remotesensing/special_issues/misr_rs).

The primary mission of the MISR instrument is to acquire reflected radiance measurements while overflying the day side of the Earth, for the purpose of quantitatively characterizing the state and evolution of the Earth's atmosphere (clouds and aerosols) and surface (vegetation and soils over land, or water properties over oceans), including the cryosphere (glaciers, ice caps and sea ice). Specifically, its combined multi-angular and multi-spectral capability has proven very useful to investigate land surface processes, especially when coupled with in situ observations (field campaigns) or other sources of environmental data (e.g., Armston et al. (2007); Chopping et al. (2008); Pisek and Chen (2009); Wei and Fang (2016); Liu et al. (2018); Mahlangu et al. (2018); Van den Hoof et al. (2018)). All MISR data products are archived at and distributed from the NASA Atmospheric Science Data Center (ASDC) in Hampton, Virginia.

It turns out that the Level 1B2 Georectified Radiance Product (GRP) data product exhibits a variable number of missing values, especially over continental areas. For reasons that will be explored in the next section, this issue may happen randomly anywhere and at any time, but occurs mostly whenever the instrument is switched between its default GLOBAL MODE and the occasional LOCAL MODE of operation. These events often perturb most spectral bands of a camera, and more than one camera.

Missing lines are detected as part of the Level 1 processing and marked with the appropriate code indicating that measurements were lost. All data products subsequently derived from these sources are of course unavailable too.

These events imply significant consequences for land surface products, which often rely on multiple spectral bands, e.g., vegetation indices, or on the whole range of cameras and bands, e.g., in assessing advanced products like surface albedo and the Fraction of Absorbed Photosynthetically Active Radiation (FAPAR), for instance. Such data losses can be crippling to the
60 investigation, especially if they occur systematically in space and time, or at the very place and time when a field campaign is taking place.

A procedure is thus required to minimize the number of missing values in MISR data files, and to maximize the spatial coverage of derived land surface products. The purpose of this paper is to report on the prevalence of this problem, to describe a proposed solution, and to show the performance of the suggested approach through concrete examples.
Section 2 discusses the origin and distribution of missing data in MISR L1B2 GRP data files. Section 3 describes the various input data files required, as well as the proposed algorithm, while Section 4 exhibits concrete results. Finally, Section 5 explores the effectiveness of that algorithm.

## 2 Problem statement and motivation

The primary MISR data, from which all surface products are derived, are the L1B2 Georectified Radiance Products (GRP),
which contain, in particular, the terrain-projected product of interest here. The latter provides calibrated multi-angular and multi-spectral observations of the planetary reflectance measured by the instrument at the nominal 'Top of the Atmosphere' (ToA), projected to a Space-Oblique Mercator (SOM) grid and geo-referenced and corrected for terrain relief effects using a digital elevation model.

### 2.1 Global and Local Modes of operation

L1B2 GRP radiance measurements in each of the nine cameras and four spectral bands are acquired by the instrument at the native spatial resolution of 275 m (across-track, for the eight off-nadir pointing cameras), or 250 m (across-track, for the nadir-pointing camera), although the latter is subsequently resampled to 275 m to provide the same sampling in all full resolution data channels (Jovanovic et al. (1999), Bull et al. (2011)). The resulting instrument-wide rate of data acquisition exceeds the downlink bandpass allotted to MISR by the Terra platform. To fit that constraint in its default GLOBAL MODE of operation,
12 of those 36 data channels (the four spectral bands of the nadir-pointing camera and the red spectral band of the eight off-nadir pointing cameras) are transmitted to the ground segment at the full, native spatial resolution, while the remaining 24 data channels (i.e., the non-red spectral bands of the off-nadir pointing cameras) are averaged on $4 \times 4$ pixel areas on-board the platform. This process achieves a 16:1 compression ratio in those data channels, and thus delivers data at the reduced spatial resolution of 1100 m. As a result, all standard MISR land products available from NASA are generated at this lower spatial
resolution.

Upon request, the MISR instrument can also be operated in LOCAL MODE (LM). When this mode is activated, each camera is allowed to transmit data in turn at the native spatial resolution of the detectors in all four spectral bands, for a limited period of time, corresponding to a scene of about 300 km along-track. LM acquisitions occur mostly over continental areas, and are requested in three broad categories of cases: (1) to systematically assess the performance of the instrument through vicarious calibration exercises over fixed, well documented sites, and throughout the mission, (2) to support field campaigns, over the areas of interest and for the duration of those exercises, or (3) to promote thematic studies, for instance over polluted urban areas, or to monitor a dynamically evolving region. About 100 sites have been identified for LM acquisition over the first 20 years of operation of MISR, with periods of observation ranging from a few weeks to the entire duration of the mission. The number of LM acquisitions per ORBIT is a mission design constraint, as each one requires the specific re-programming of the instrument and leads to larger data transfers to the ground segment. In practice, only one LM acquisition is allowed per ORBIT on an operational basis. At the time of writing (mid-March 2020), LOCAL MODE acquisitions were undertaken roughly three to eight times a day (out of 14 opportunities). The ASDC maintains a table, updated daily, reporting on the success of LM acquisitions (see https://l0dup05.larc.nasa.gov/public/cgi-bin/DUE/ecs_LocalMode_history_PR.cgi).

During the initial pre-processing phase, the original analog radiance measurements acquired by the instrument's cameras are converted to 16-bit unsigned integers, where the first (most significant) 14 bits contain the scaled radiance itself and the last 2 bits constitute the Radiometric Data Quality Indicator (RDQI), an indicator of the estimated reliability of the data item. The floating point radiance is obtained by unpacking those last 2 bits and multiplying the resulting value by the scale factor that is included in the data file. There is also a one-to-one relation between the radiance and the non-dimensional bidirectional reflectance factor (denoted Brf in this paper; (Bull et al., 2011, p. 73)):

$$\text{Brf} = [(\pi \times D^2)/(W \times \cos\theta_s)] \times \text{Rad} \tag{1}$$

where $D$ is the approximate distance in astronomical units between the center of the Earth and the center of the Sun at the time MISR observes the first valid (non-fill) pixel in the swath (this information does not vary with spectral bands but is replicated across bands for convenience (Bull et al., 2011, p. 60)); $W = E_0^{std}(b)$ is the band specific weighted standardized solar irradiance in W m$^{-2}$ $\mu$m$^{-1}$ (Bull et al., 2011, p. 65–68); $\theta_s$ is the solar zenith angle, and Rad stands for the floating point radiance value, in W m$^{-2}$ sr$^{-1}$ $\mu$m$^{-1}$.

The on-board computer that manages the measurements and performs the GLOBAL MODE (GM) spatial averaging may occasionally be overwhelmed by the high acquisition data rate. This appears to occur randomly at a low but unpredictable frequency. However, the number of missing data values increases by at least an order of magnitude whenever the instrument is switched from the default GLOBAL MODE to the LOCAL MODE (LM) of operation, and conversely. A procedure is thus required to replace those missing values, and it must be designed to work effectively everywhere and all the time.

## 2.2 Detecting missing values

The first step in this investigation consists of documenting the prevalence of this problem, which implies distinguishing missing values due to this on-board computer glitch from other causes. As explained above, radiance values are represented in the L1B2

GRP data files by 16-bit unsigned integers, where the last 2 bits are the RDQI. Table 1 describes the meaning attached to the values of this quality indicator (Bull et al., 2011, p. 71).

**Table 1.** Meaning of the 2-bit RDQI values in L1B2 GRP data files.

| Decimal | Binary | Meaning |
| --- | --- | --- |
| 0 | 00 | Within specifications ('good') |
| 1 | 01 | Reduced accuracy ('fair') |
| 2 | 10 | Not usable for science ('poor') |
| 3 | 11 | Unusable for any purpose ('missing') |

The algorithm used to assign the RDQI value to each pixel is described in detail in Jovanovic et al. (1999). Practical experience with this data product indicates that a RDQI value of 1 refers to either a minor increase in the uncertainty of the radiance value or a possible slight mis-registration of the pixel, while a value of 2 indicates a larger radiance uncertainty, for instance in the presence of sun glint reflected from a still water surface. An RDQI value of 3 can be assigned to a terrain-projected L1B2 GRP value in four different cases, and the radiance value itself indicates why those values are unusable:

1. when the location of a pixel on the ground is obscured by topography for the specified camera, an issue that mostly affects off-nadir pointing cameras: its value is coded as the unsigned 16-bit integer 65511, equivalent to a scaled 14-bit radiance of 16377,

2. when the location of a pixel on the ground lies outside of the swath width of the camera, which occurs on the western and eastern edge of each BLOCK: its value is coded as the unsigned 16-bit integer 65515, equivalent to a scaled 14-bit radiance of 16378,

3. when the location of a pixel at the Earth's surface belongs to a BLOCK covering an ocean region only: its value is coded as the unsigned 16-bit integer 65519, equivalent to a scaled 14-bit radiance of 16379, or

4. when the radiance estimated for a location is deemed unusable for a reason other than orography, edge or ocean: its value is coded as the unsigned 16-bit integer 65523, equivalent to a scaled 14-bit radiance of 16380: this is a 'missing' value in the context of this paper.

Consequently, the RDQI is an indicator of missing values in the sense described earlier only if it is associated with an unsigned 16-bit radiance value of 65523.

Figure 1 exhibits a simple example: in this case, a couple of lines have been dropped from the blue spectral band in BLOCK 110 of the DF camera in the LOCAL MODE bidirectional reflectance factor data file. In this context, a BLOCK is a segment of data that covers a ground area of about 563.2 km across-track by 140.8 km along-track, which contains usable data over roughly 380 km across-track, as well as fill data on both edges. Figure 2 of Verstraete et al. (2020) shows the geographical

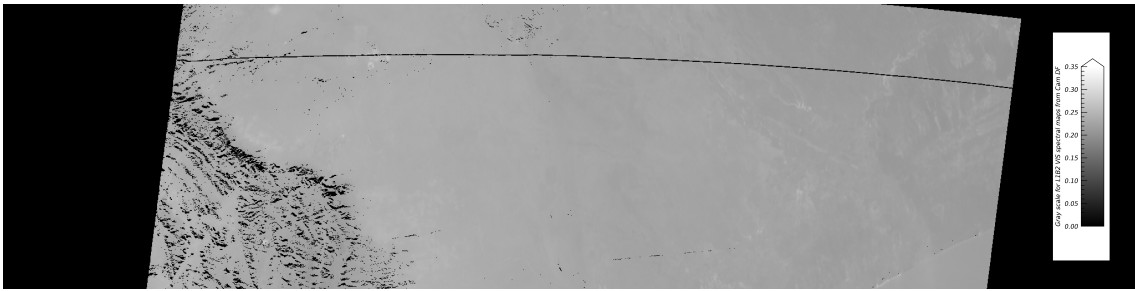

**Figure 1.** Example of missing lines (in this case, two consecutive lines) in the L1B2 terrain-projected LOCAL MODE bidirectional reflectance factor data for the blue spectral band of the DF camera, for PATH 168, ORBIT 68050, and BLOCK 110, covering an area located in eastern South Africa. The linear grey scale, shown on the right side of the frame, assigns Brf values of 0 to black and Brf values of 0.35 or higher to white. The black spots on the western side of the map are due to pixels that are not observed at this slanted view angle due to the local topography. North is approximately pointing towards the top of the image. The total size of the BLOCK area is 563.2 km across-track (roughly west-east) by 140.8 km along-track (roughly north-south), while the parallelogram-shaped ground area inside the BLOCK is about 380 km across-track.

location of this BLOCK in Southern Africa. These missing lines tend to occur simultaneously in all spectral bands, but because the spectral detectors are located next to each other on the focal plane of each camera, the corresponding locations of these missing values on the ground, and therefore in the images, are slightly displaced. As a result, products that combine multiple spectral bands from the same camera inherit all missing lines from each of the individual bands. And of course, a missing line in a LOCAL MODE data also implies a missing line in the corresponding GLOBAL MODE data, with an added side effect: the missing lines in the non-red spectral channels of the off-nadir cameras show up with a width of 1100 m (or multiples thereof). Figure 2 exhibits the RGB image for the GLOBAL MODE acquisition corresponding to the case shown in Figure 1 (top frame), as well as the corresponding map of the NIR spectral band for the same camera.

## 2.3 Counting missing and poor values

An examination of the MISR L1B2 GRP Terrain-Projected GLOBAL MODE (GM) Radiance files for BLOCKs 110 to 113 of PATHs 168 to 170, from the start of the mission onwards, revealed that millions of pixel values went missing (RDQI of 3 and scaled radiance of 16380) in Global as well as LOCAL MODE acquisitions.

For instance, Figure 3 exhibits the logarithm in base 10 of the number of missing pixels in the four spectral bands of the nine cameras for BLOCK 110 of all available ORBITs of PATH 168 between March 24, 2000 and December 23, 2018: A total of 2,143,033 pixel values went missing. LOCAL MODE data were briefly acquired at the start of the MISR mission for the Skukuza flux tower site located in the Kruger National Park of South Africa, in support of the Safari-2000 campaign. This site, which covers an area roughly equivalent to BLOCKs 110 and 111, has subsequently been systematically acquired from December 2009 onward. The net effect of switching between GLOBAL to LOCAL MODE (and back) on the number of missing

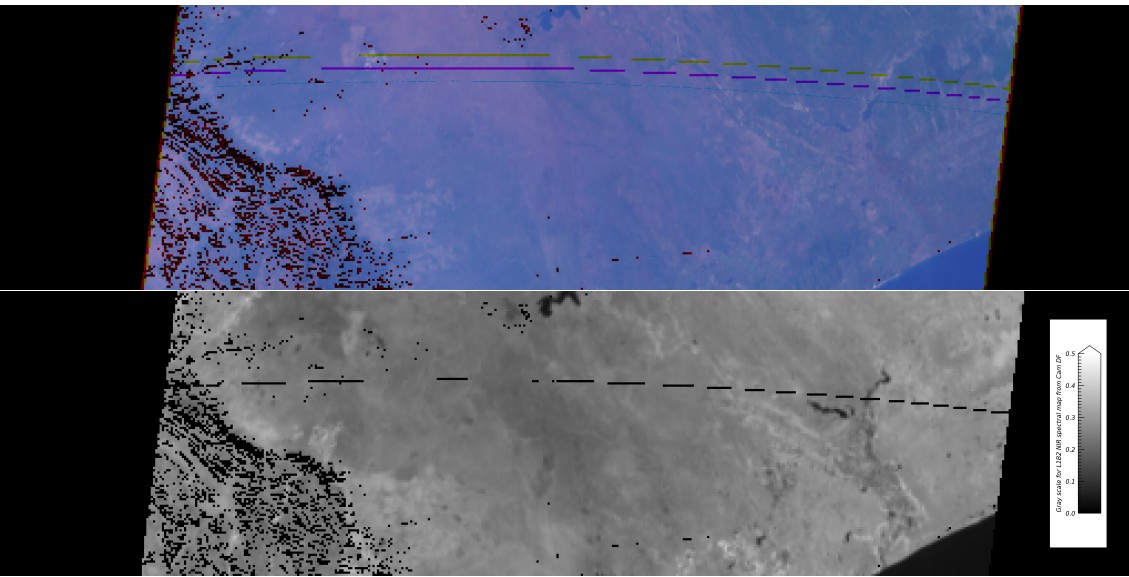

**Figure 2.** Top frame: Example of the effect of missing lines in a multispectral product, here an RGB composite of the red, green and blue spectral bands for the L1B2 terrain-projected GLOBAL MODE Brf data for the same DF camera case exhibited in Figure 1. The top (yellow), middle (magenta) and bottom (light cyan, barely visible because of the general bluish tone of the image) lines correspond to the missing data in the blue, green and red spectral bands, respectively. This image was generated by first replicating the low resolution blue and green pixels so as to match the resolution of the red band, and then creating the composite image at the full spatial resolution. Bottom frame: Map of the NIR Brf of the same DF camera. Linear dimensions are the same as in Figure 1. The three RGB spectral channels are stretched from 0.0 to 0.35 for the purpose of generating the color image, while the NIR data is stretched between 0.0 and 0.5.

values is clearly visible in this logarithmic plot: the number of missing values is about an order of magnitude larger when LOCAL MODE acquisitions are taking place.

A close inspection of the data quality in these files also revealed that missing values are often associated with 'poor' or 'fair' radiance values (with an RDQI value of 2 or 1, respectively) in their immediate vicinity. Figure 4 shows the time series of all poor values found in the four spectral bands and nine cameras for BLOCK 110 of all available ORBITs of PATH 168 between the start of the mission and December 23, 2018. Some 918,420 values fell in that category during that period: that brings the total number of missing or poor values to over 3 million for that particular PATH and BLOCK.

These findings call for multiple comments:

– While these numbers appear large (out of context), they remain tiny compared to the total number of 'good' and 'fair' pixel values (i.e., those associated with a RDQI of 0 or 1), which is of the order of $25 \times 10^9$ pixels for the same BLOCK and time period. So this problem does not affect the usability of the MISR instrument, especially for studies involving large areas or long periods of time.

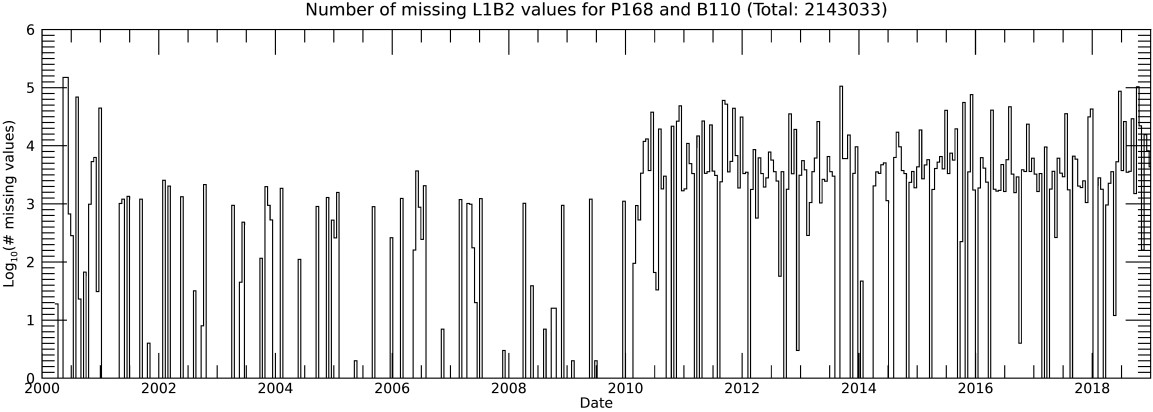

**Figure 3.** Time series of the logarithm in base 10 of the number of missing pixel values, in the four spectral bands of the nine cameras of BLOCK 110 from each available MISR L1B2 GRP ORBIT in PATH 168, between March 2000 and December 2018. This BLOCK includes the Skukuza flux tower located in the Kruger National Park of South Africa. LOCAL MODE data were briefly acquired at the start of the mission to support the Safari-2000 campaign, and then systematically from December 2009 onward. It can be seen that there is a strong relation between the number of missing values and the acquisition of LOCAL MODE data, although the latter is not the only process responsible for missing data.

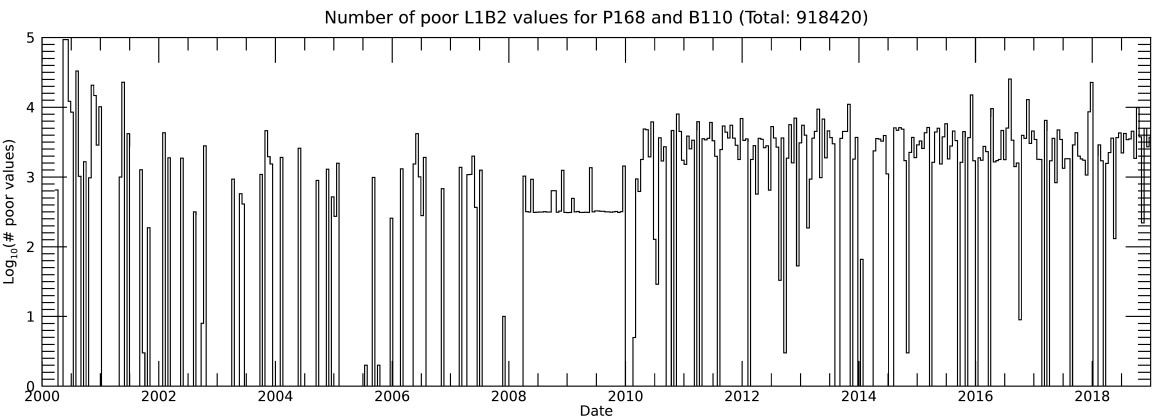

**Figure 4.** Time series of the logarithm in base 10 of the number of poor pixel values, in this case in the four spectral bands of the nine cameras of BLOCK 110 from each available MISR L1B2 GRP ORBIT in PATH 168, between late March 2000 and December 2018 (the same BLOCK as in Figure 3). See the text for a discussion of the similarities and differences between those two time series.

– However, for the technical reasons described above, missing values are often concentrated in geographical areas where LOCAL MODE acquisitions take place. Hence, if a site is systematically acquired in LOCAL MODE over a substantial

period of time, then that region tends to be more affected by this problem than other areas. This, in turn, implies that local studies, field campaigns or projects that occur precisely in those areas and time periods may be severely impacted while similar investigations elsewhere may be relatively unaffected.

- There are somewhat fewer poor values than missing values in general, but there is also less difference in the peak number of poor values with or without LOCAL MODE acquisitions. This is because measurements affected by sun glint are typically marked as poor. This natural phenomenon occurs systematically and may affect a camera irrespective of the LOCAL or GLOBAL MODE of acquisition, especially whenever the observed scene contains substantial water bodies.

- Nevertheless, as was the case for L1B2 radiance values, the number of poor values also increased by about an order of magnitude whenever the Skukuza site was systematically observed in LOCAL MODE.

- Figure 4 also reveals that a significant number of radiance measurements were assigned an RDQI of 2 between about March 2008 and the end of 2009, for reasons that remain to be investigated but that probably involve changes in the software processing system, rather than the performance of the instrument.

The study of Mahlangu et al. (2018) provides a good example of the crippling effect of these missing and poor data on a particular investigation: An airborne LiDAR field campaign took place in and around the Skukuza site in April 2012. The aim of that study was (1) to document the structure of vegetation using the LiDAR as a high spatial resolution reference data over a limited area, and (2) to explore whether a relation between these airborne measurements and MISR-derived products could help describe those ecosystem properties over a larger region. It relied on products to characterize the plant canopy generated by the MISR-HR processing system (Verstraete et al. (2012), Version 1.04-5) for the closest available acquisition, namely PATH 168, ORBIT 65487 and BLOCK 110, acquired on 10 April 2012. The MISR-HR processing system is a set of algorithms designed to regenerate best estimates of the MISR data at the full spatial resolution in all 36 data channels. Figure 5, demonstrates the problem: the spatial coverage of the Leaf Area Index (LAI) product over the BLOCK was significantly reduced by clouds (screened out on square areas of 17.6 km by 17.6 km by the software and data available at the time that paper was generated) and the remaining clear area was very much contaminated by missing data lines in all four spectral bands of two different cameras (AF and CA).

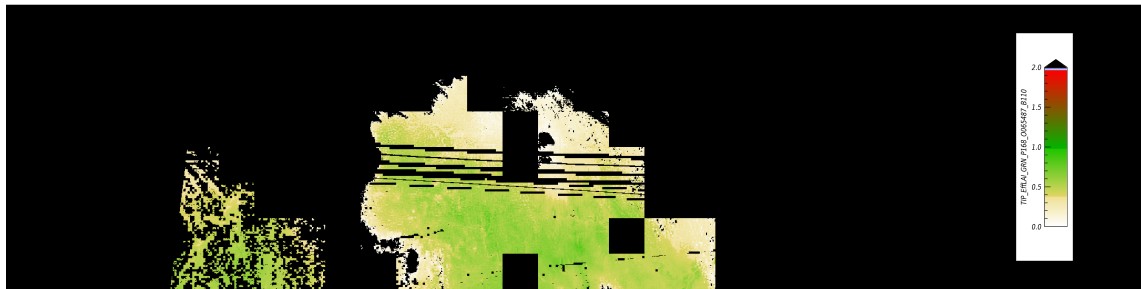

**Figure 5.** Example of the effect of missing lines in a derived product, here the Leaf Area Index (LAI) for the particular date and time when an airborne field campaign was taking place in the Kruger National Park of South Africa (the same PATH 168, ORBIT 65487, BLOCK 110 as for Figures 1 and 2; see text for details). A large fraction of the MISR BLOCK is unavailable (depicted in black), due to the prevailing cloud cover on that particular date as well as the presence of missing lines. The color gradient (white, yellow, green, red) maps the LAI range [0.0, 2.0].

In summary, MISR L1B2 data losses can occasionally occur anywhere and at any time, and will involve tens of thousands of missing values over a period of 10 years. However, in the neighborhood of a site that is systematically monitored in LOCAL MODE, the data loss can amount to multiple millions of missing values over the same period. These losses will be of little consequence for climatological studies, but may cripple local applications.

To address this issue and avoid serious contamination of subsequent analyses and products generated by the MISR-HR processing system (Verstraete et al., 2012), the radiance values that are missing, and optionally those associated with an RDQI of 2, should be replaced by best estimates of what the original measurements would likely have been. The next sections describe the proposed solution to this problem, exhibit some results and document the performance of the algorithm.

## 3 Materials and Methods

### 3.1 MISR data

The MISR mission is run by a group of engineers and scientists at NASA's Jet Propulsion Laboratory (JPL) in Pasadena, California, supported by an international team of researchers, while the data generated by this instrument and the standard products derived from the raw measurements are processed and permanently archived at NASA's Atmospheric Science Data Center (ASDC) in Hampton, Virginia. As is the case for other Earth Observation missions, all MISR data products are freely available to anyone interested, either through customized orders, or through bulk downloads from the Data Pool at https://eosweb.larc.nasa.gov/project/misr/misr_table.

Three standard data sets are of direct relevance for the current purpose: the AGP and two out of the six parameter sets of the Level 1B2 Georectified Radiance Product (GRP): the terrain-projected Top of Atmosphere (ToA) radiance GLOBAL MODE parameter and the Radiometric Camera-by-Camera Cloud Mask (RCCM) (Bull et al., 2011, p. 56).

### 3.1.1 The AGP dataset

The Ancillary Geographic Product (AGP) is the master reference dataset containing essential information on the latitude and longitude of BLOCK locations, as well as ancillary data such as the expected distribution of land masses and water bodies (in particular oceans) (Bull et al., 2011, p. 14). This primitive land cover map actually recognizes 7 different types of "surface features" (Bull et al., 2011, p. 210):

- – 0: Shallow Ocean

- – 1: Land

- – 2: Coastline

- – 3: Shallow Inland Water

- – 4: Ephemeral Water

- – 5: Deep Inland Water

- – 6: Deep Ocean

The spatial distribution of these feature types is fixed throughout the mission: it is not intended as a genuine, detailed land cover map, but as a coarse land-sea partition, with a few more features. In the current context, categories 0, 5 and 6 are merged into a water class, and the remaining categories are assigned to the land class. The results described in this paper are based on the AGP product Version `F01_24`.

### 3.1.2 The Terrain-projected ToA radiance Global Mode parameter set

This parameter set is the primary MISR radiance product at the nominal Top of the Atmosphere (ToA): all higher level products, including the RCCM described in the next subsection, are derived from an analysis of those calibrated and geolocated measurements. In the default GLOBAL MODE of operation, two versions of that parameter set are generated: one in which all measurements are projected on the World Geodetic System 1984 (WGS84) reference ellipsoid, primarily used in applications over oceans, and one where measurements are projected on a Digital Elevation Model (DEM), primarily used over land. This paper is concerned exclusively with this latter projection. All results described in this paper are based on L1B2 Terrain-projected Georectified Radiance Product Version `F03_0024`.

### 3.1.3 The RCCM parameter set

The Radiometric Camera-by-camera Cloud Mask (RCCM) provides a separate cloud mask for each of the nine MISR cameras. This standard data set is described in detail in the Level 1 Cloud Detection Algorithm Theoretical Basis Document (ATBD) (Diner et al., 1999b), which can be downloaded from https://eospso.gsfc.nasa.gov/atbd-category/45, and in subsequent papers

such as Zhao and Di Girolamo (2004) and Yang et al. (2007). All results described in this paper are based on RCCM Version F04_0025.

Over land, these camera-specific cloud masks are derived from the L1B2 radiance data described in the previous subsection, using the red and the near-infrared (NIR) spectral bands of each camera. Hence, whenever there are missing data in those spectral bands, there will be missing data in the RCCM product too. However, a procedure to replace missing data in those cloud masks has recently been designed and is fully documented in Verstraete et al. (2020).

### 3.2   Correlation between the target and the sources

A practical solution to the problem described in Section 2 above (i.e., restoring data when there are missing lines) is made possible by two findings: (1) as shown earlier, the missing pixels appear in different locations in different spectral channels of the affected camera, and (2) in most cases, it is possible to find, for each data channel affected by missing or poor data, one or more data channels (among the other 35 simultaneously acquired, i.e., for the same MISR PATH, ORBIT and BLOCK) that exhibit a very high correlation with the one in need of an update. In the rest of this paper, the data channel that is being repaired is called the *target*, and the other 35 data channels of the same data acquisition are called the *sources* (or predictors).

The Pearson correlation coefficients (CC) between all pixel values that are valid (i.e., excluding all values with an RDQI value of 2 or 3) and common between the target and each of the 35 source data channels are computed. In each of these 35 cases, the root mean square difference (RMSD) between the data channel values, the best linear fit between the source and the target, and the $\chi^2$ value, i.e., a measure of the dispersion of the data points around that linear fit function, are also calculated. This set of statistics is then ranked in decreasing order of correlation coefficient value. It turns out that one or more source data channels are usually highly correlated (CC larger than 0.9) with the target data channel, thereby providing reliable means to replace the missing values in the target on the basis of non-missing values in the most promising source data channel.

This approach is first demonstrated on the MISR GRP GLOBAL MODE data for PATH 168, ORBIT 68050, BLOCK 110, camera DF exhibited in Figure 2. Table 2 shows the statistics describing the relations between the four target spectral bands of the DF camera and the best (i.e., the most promising, in terms of the Pearson CC) identified sources (in this case the same spectral channels of the CF camera), using all valid pixels that are common to both data channels. The $\chi^2$ value for the red band (or indeed for any high resolution data channel) is much larger than for the other GLOBAL MODE data channels because they contain about 16 times more data points.

**Table 2.** Correlation statistics between the radiance in the four spectral bands of the target (Tgt) camera DF and the radiance in the best correlated source (Src) data channels (first attempt), using all common and valid values for the GLOBAL MODE measurements acquired in PATH 168, ORBIT 68050, and BLOCK 110. The columns entitled 'No. Pix.', 'RMSD', 'PCC' and '$\chi^2$' provide the number of valid measurements common to both data channels, the root mean square difference between the data channel values, the Pearson correlation coefficient between them and the value of that statistics for the linear fit between those data channels, respectively.

| Tgt. Cam. | Tgt. Band | Src. Cam. | Src. Band | No. Pix. | RMSD | PCC | $\chi^2$ |
|---|---|---|---|---|---|---|---|
| DF | Blue | CF | Blue | 45779 | 16.32 | 0.951 | 71875 |
| DF | Green | CF | Green | 45941 | 5.283 | 0.969 | 53560 |
| DF | Red | CF | Red | 762392 | 2.409 | 0.977 | 2.23E+06 |
| DF | NIR | CF | NIR | 46186 | 1.741 | 0.989 | 129323 |

Figure 6 shows the scatterplots and the linear fits between the four target spectral bands of the DF camera case mentioned in Table 2 and the best predictor source data channels for the selected BLOCK.

While the correlations are highly significant, in part due to the large number of pixel values, the shape and the dispersion of the cloud of data points often suggests that a single linear equation may not always be optimal to predict the target values across the entire BLOCK. This is largely due to the fact that the scene may include rather different geophysical media, which follow possibly different statistical relations as far as the correlations between data channels are concerned.

To explore this avenue, the sets of points used in these computations were partitioned into three categories: clear land masses, clear water bodies, and cloud fields. The geographical distribution of land masses and water bodies was extracted from the MISR surface cover type map contained in the static AGP file described earlier, for the specified PATH and BLOCK. The cloud information was retrieved from the Radiometric Camera-by-Camera Cloud Mask (RCCM) data product, updated as explained in Verstraete et al. (2020). Statistics, including the Pearson correlation coefficients and the best linear fit equations, were generated for each of these three categories. The gain in pursuing this approach is demonstrated in Figure 7, which compares the best fit statistics while processing the NIR spectral band of Camera AA for the same scene as above, when all valid values available in the BLOCK are used for establishing the statistics (left panel), and when only those values attributable to clear land masses are used (right panel). The RMSD, the CC and the $\chi^2$ statistics are significantly improved by focusing on the clear land masses only.

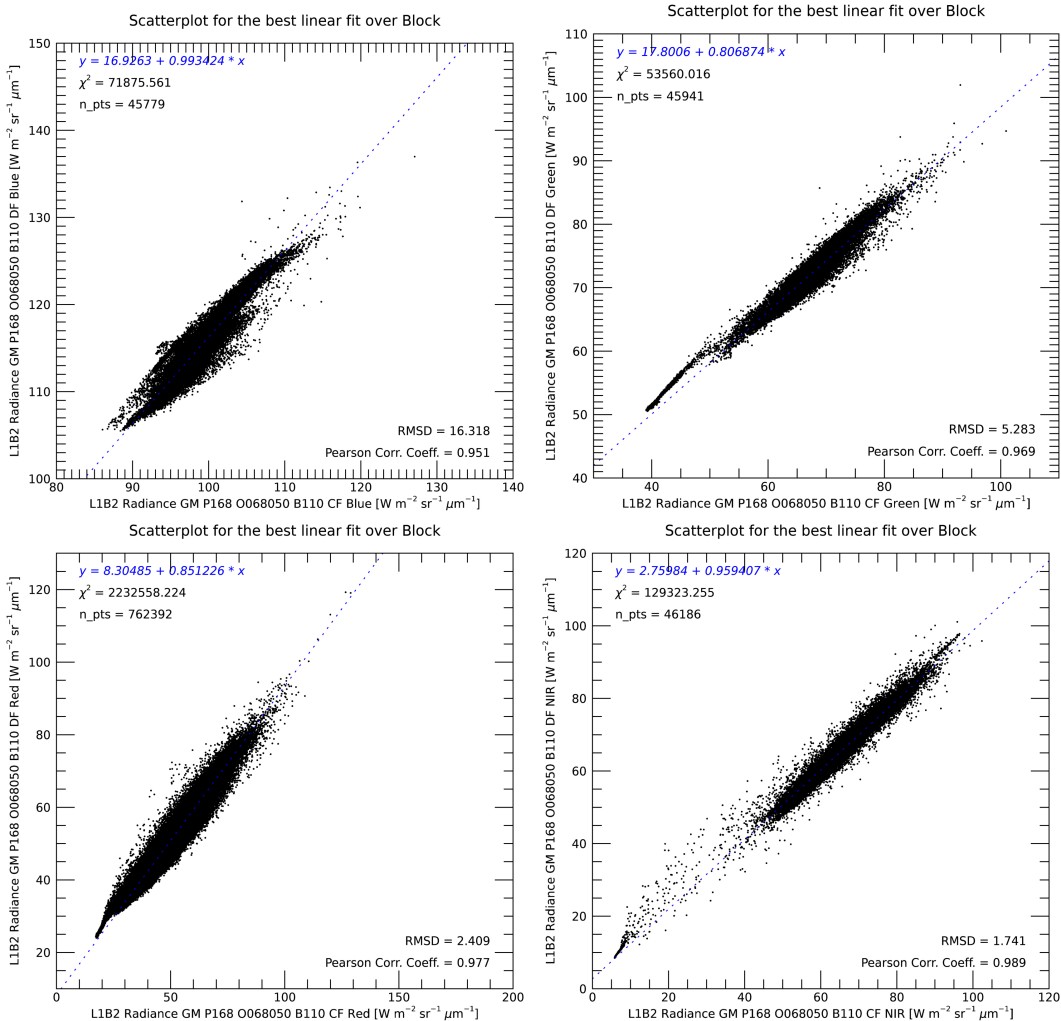

**Figure 6.** Scatterplots and best linear fit relations to predict the radiance values of the four spectral bands of the target DF camera for the GLOBAL MODE measurements of PATH 168, ORBIT 68050, and BLOCK 110, using all common valid values. In this case, the best predictors for the measurements by the DF camera were those acquired by the CF camera in the same spectral bands as the targets (Top left panel: blue, top right panel: green, bottom left panel: red, and bottom right panel: NIR). Note that the ranges of values are different for each plot in order to optimize the view of the data.

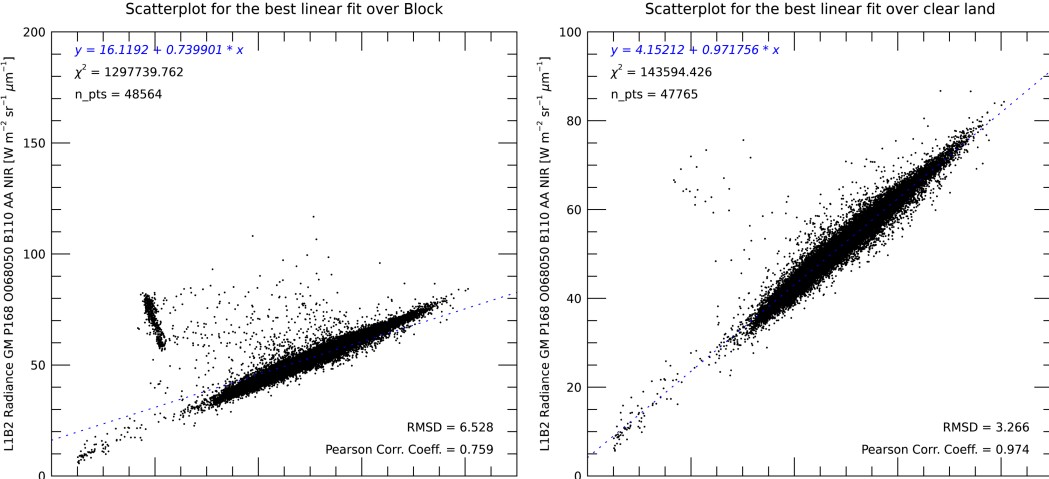

**Figure 7.** Scatterplots and best linear fit relations to predict the radiance values of missing data in the NIR spectral band of the AA camera, for the same GLOBAL MODE measurements of PATH 168, ORBIT 68050, and BLOCK 110 as in Figure 6. The left panel shows the results obtained when all common valid values throughout the BLOCK are considered, while the right panel shows the statistical results applicable only to the clear land masses. The Pearson correlation coefficient, the RMSD and the $\chi^2$ statistics are definitely improved in this latter case, as a group of more than 800 pixel values, identified as clouds, is treated separately.

Table 3 shows the statistical results for the same cases as before (Table 2), this time reporting values for clear land masses only. While the correlation coefficients are roughly equivalent, despite the somewhat smaller number of points involved, the dispersion statistics (RMSD and $\chi^2$) are again improved. Figure 8 exhibits the corresponding scatterplots and the best linear fit functions for clear land masses: as expected, these compare favorably to the results shown in Figure 6, since this particular BLOCK is essentially cloud-free.

**Table 3.** Correlation statistics between the four spectral bands of the target camera DF and the best correlated source data channels (first attempt), using all common and valid clear land values for the GLOBAL MODE measurements acquired in PATH 168, ORBIT 68050, and BLOCK 110. The column headers are identical to those in Table 2.

| Tgt. Cam. | Tgt. Band | Src. Cam. | Src. Band | No. Pix. | RMSD | PCC | $\chi^2$ |
|---|---|---|---|---|---|---|---|
| DF | Blue | CF | Blue | 44729 | 16.27 | 0.957 | 63244 |
| DF | Green | CF | Green | 44866 | 5.132 | 0.962 | 49093 |
| DF | Red | CF | Red | 744596 | 2.242 | 0.974 | 2.15E+06 |
| DF | NIR | CF | NIR | 45066 | 1.652 | 0.982 | 120534 |

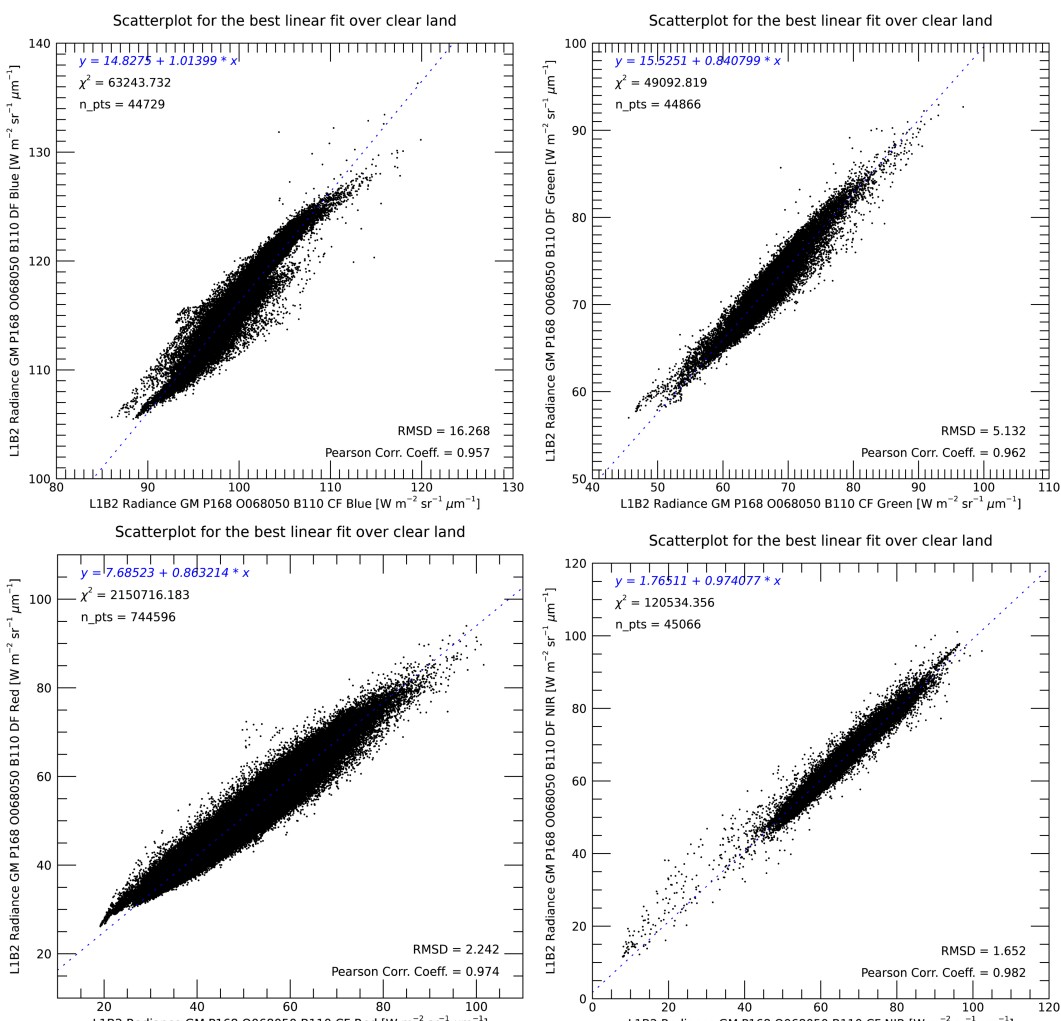

**Figure 8.** Scatterplots and best linear fit relations to predict the clear land surface values of the four spectral bands of the DF camera for the GLOBAL MODE measurements of PATH 168, ORBIT 68050, and BLOCK 110. In this case, the best predictors for the measurements by the DF camera were those acquired by the CF camera (Top left panel: blue, top right panel: green, bottom left panel: red, and bottom right panel: NIR). Note that the ranges of values are different for each plot in order to optimize the view of the data.

### 3.3 Replacement algorithm

These findings suggest that missing and optionally poor L1B2 radiance values could be replaced by reconstructed estimates based on the actual measurements obtained quasi simultaneously in other data channels (spectral bands or cameras), for the same geographical location. This Section outlines the sequence of steps undertaken to achieve this goal, in a schematic manner intended to also serve as a guide to the IDL function `fix_l1b2.pro` which implements these concepts in the open source IDL processing software Version 2.2.0 available from GitHub.

This function takes as input arguments pointers to the scaled radiances (with the RDQI attached), the unscaled radiances, the Brf and the RDQI data buffers for the specified MISR LOCAL or GLOBAL MODE, PATH, ORBIT and BLOCK, which must have been loaded on the heap prior to the call. This ensures a fast access to those data fields, as they will need to be read multiple times during the processing. An initial suite of preliminary steps is then undertaken to set the stage:

- The IDL function offers the option of mapping the L1B2 product before processing, to be able to show how the results of the processing differ from the original MISR data.

- The next step consists in generating the masks for clear land masses, clear water bodies and cloud fields, to partition all valid radiance values found in the BLOCK into those three categories. Although statistics are in fact computed for clear water bodies and cloud fields, they are not reported here, as the land areas are of primary interest.

- An optional main log file may be created to keep track of the progress and provide information on intermediary results, for instance for diagnostic purposes.

- The function `pre_fix_l1b2.pro` is then called to locate, within each of the 36 data channels, the numbers and positions of the poor and missing data values. If any are found, the function `best_fits_l1b2.pro` is activated to compute the optimal statistics and the best linear fit functions to replace those poor or missing values, and to optionally generate a separate log file for each case. Information on progress is also added to the main log file if it has been requested in the previous step.

At that point, the code relies on two closely related functions, `fix_poor_l1b2.pro` and `fix_miss_l1b2.pro`, to replace poor (if required) and missing L1B2 values, respectively. The core of the processing is outlined in the following pseudo-code listing:

```
FOR each camera
    FOR each spectral band
        access the MISR L1B2 data for the target
        retrieve the number and location of poor or missing values
        skip to the next camera/band combination if no values need replacement
        WHILE the maximum number of attempts has not been reached
            IF poor or missing values remain in the land/water/cloud target
                record the current best source and correlation statistics
                access the MISR L1B2 data for the current source
                upscale or downscale the source data to match the target resolution
                FOR each target value that needs replacement
                    record the radiance value at the same location in the current source
                    ensure that this source value is valid (i.e., not missing or poor itself)
                    apply the best linear fit equation for this attempt
```

```
                    replace the poor or missing value in the target by this new estimate
                    update the RDQI for that new value to 1B
              ENDFOR
update the number and location of remaining missing (or poor) values
          ENDIF
          point to the next best source and correlation statistics
        ENDWHILE
      ENDFOR
ENDFOR
```

If poor values are replaced, the function `fix_poor_l1b2.pro` also optionally generates statistics and scatterplots of the updated versus the original values.

The `WHILE` loop over `attempts` is required because in some cases the best source data channel to replace values in the target data channel happens to also have missing values in the very same location as the target. In those cases, the next best

data channel must be considered. The maximum number of attempts (i.e., the maximum number of best source data channels to consider) is specified as an argument to the function. Practical experience and considerations indicate that this value should generally be set to no more than `4`: smaller values may result in non-trivial numbers of missing or poor values not being replaced, while larger values may result in the use of source data channels that are less well correlated to the target, and therefore result in proposing less reliable replacement values. However, see Subsection 4.3 below for an exceptional case that

required 8 iterations to replace most missing values. Note also that replaced values are always assigned an RDQI of `1`.

## 4    Results

The following exhibits will show graphically the outcome of this process, for the cases discussed in Section 2 above.

### 4.1    Global Mode, Path 168, Orbit 68050, Block 110, Camera DF

The particular case exhibited earlier in Figure 2, where four spectral bands of the DF camera exhibit missing and poor data

values, is examined first.

Figure 9 shows the outcome of the process where each and every missing value in the original target data channels is replaced by a value estimated by retrieving the valid bidirectional reflectance factor values retrieved from the source data channels indicated in Table 3 for the very same geographical location, and inserted into the linear fit equations exhibited in Figure 8. The top frame is an RGB composite and the bottom frame shows the map of the corresponding NIR spectral band.

These frames can be compared to those exhibited in Figure 2.

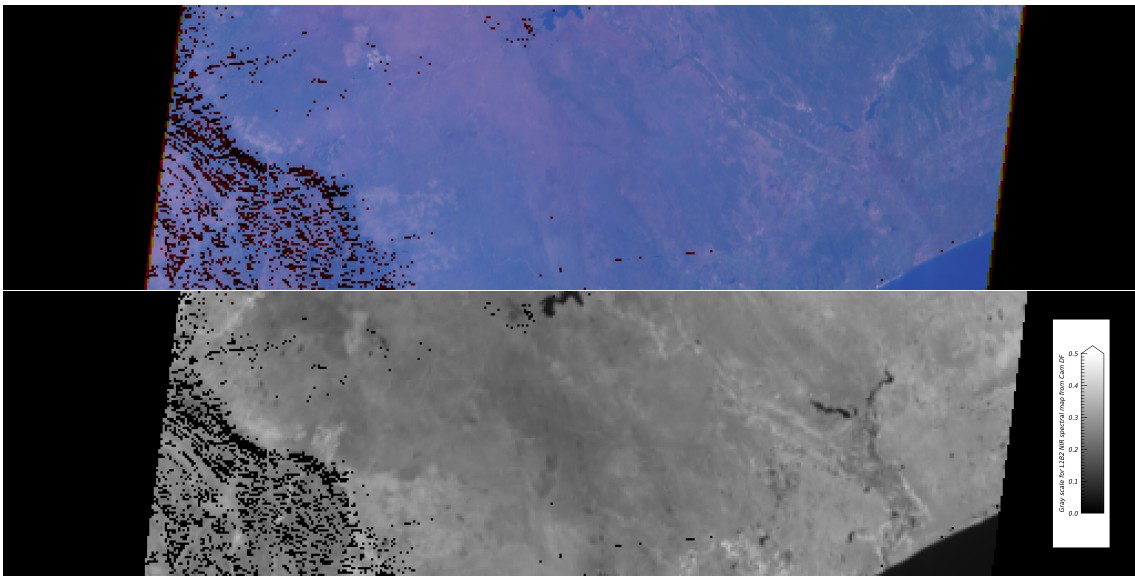

**Figure 9.** Top frame: RGB composite map of the red, green and blue spectral bands for the L1B2 terrain-projected GLOBAL MODE bidi-rectional reflectance factor data file for the same DF camera case exhibited in Figure 2. Bottom frame: Map of the NIR spectral band for the same DF camera. All missing values in the original data files have been replaced by estimates as described in the text. Linear dimensions are the same as in Figure 1, and all three spectral channels are stretched from 0.0 to 0.35 for the purpose of generating the color image; the gray scale used to stretch the NIR data for the bottom frame assigns black to reflectance values of 0.0 and white to values of 0.50 or higher.

## 4.2   Global Mode, Path 168, Orbit 65487, Block 110, Camera CA

The case exhibited earlier in Figure 5 is examined next. Table 4 summarizes the correlation statistics between the four spectral bands of the CA camera, all affected by missing values, and the best sources to estimate replacement values, considering only the valid values in clear land masses that are common to the target and the source data channels.

**Table 4.** Correlation statistics between the four spectral bands of the target camera CA and the best correlated source data channels, using only common and valid clear land values for the GLOBAL MODE measurements acquired in PATH 168, ORBIT 65487, and BLOCK 110. The column headers are identical to those in Table 2.

| Tgt. Cam. | Tgt. Band | Src. Cam. | Src. Band | No. Pix. | RMSD | PCC | $\chi^2$ |
|-----------|-----------|-----------|-----------|----------|-------|-------|----------|
| CA | Blue | CA | Green | 17591 | 22.48 | 0.966 | 156441 |
| CA | Green | CA | Blue | 17591 | 22.48 | 0.966 | 119634 |
| CA | Red | DA | Red | 268707 | 10.47 | 0.955 | 1.14E+07 |
| CA | NIR | DA | NIR | 15613 | 6.866 | 0.969 | 58113 |

Figure 10 exhibits maps of the RDQI for those four data channels of the CA camera. The relative displacement of missing lines between the spectral bands and the clustering of poor values around those lines are clearly observable: these findings have been widely confirmed in all cases investigated.

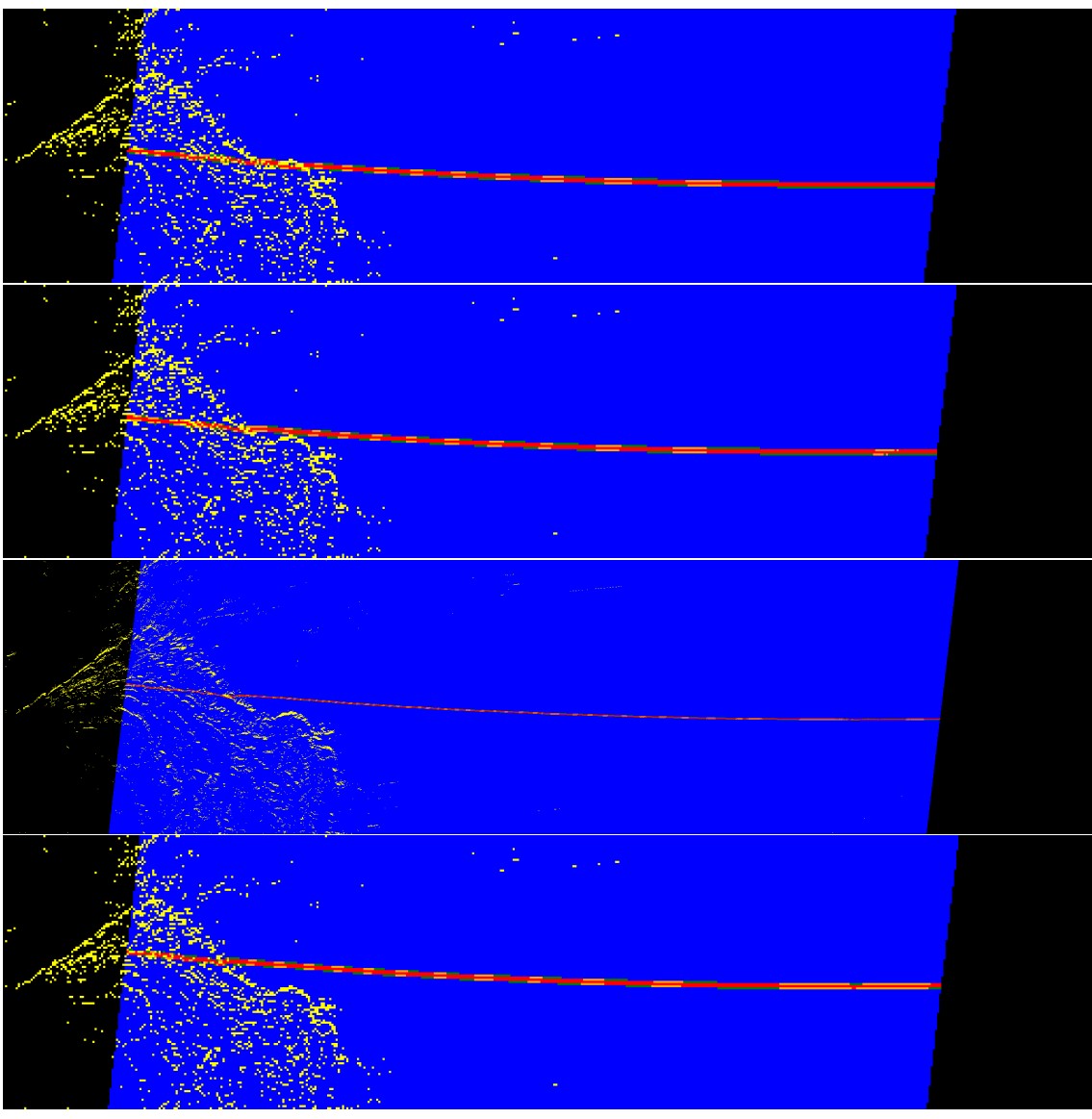

**Figure 10.** Spatial distribution of RDQI values for Global Mode, Path 168, Orbit 65487, Block 110, Camera CA, in the four spectral bands: Blue, Green, Red, and NIR from top to bottom. Color coding: Black: Edge pixels; Yellow: pixels obscured by the local topography; blue: RDQI = 0, green: RDQI = 1; orange: RDQI = 2; red: RDQI = 3.

Note that the best predictor for missing and poor values in the CA/Blue data channel is the CA/Green data channel, and conversely. Since the RMSD and the Pearson correlation coefficient are symmetric with respect to their inputs, the statistics

are identical in these cases. However, the CA/Green data points fit a little bit better than the best linear function (the $\chi^2$ for the function predicting CA/Green is somewhat lower than that for CA/Blue). The best predictors for the CA/Red and CA/NIR data channels are provided by the DA camera, in the corresponding spectral bands.

The scatterplots for these 4 cases are exhibited in Figure 11. It can be seen that the best fit function to predict the CA/Red data channel is not as good as for the other data channels, although the Pearson correlation coefficient is still 0.836. These

375 statistics are of course also affected by the fact that there are quite a lot fewer valid data values available to estimate the missing and poor pixels, compared to the earlier case of ORBIT 68050.

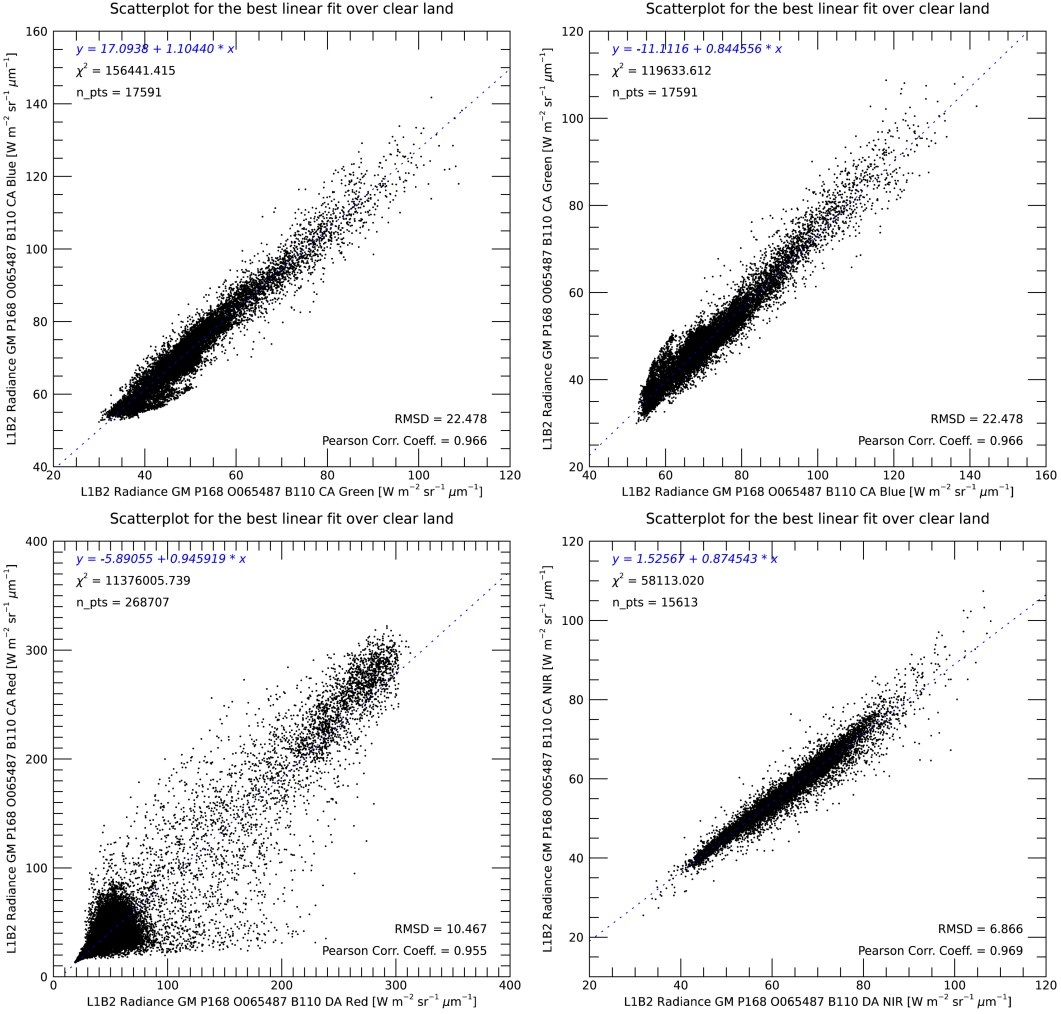

**Figure 11.** Scatterplots and best linear fit relations to predict the values of the four spectral bands of the target CA camera for the GLOBAL MODE measurements of PATH 168, ORBIT 65487, and BLOCK 110. Top left panel: Blue, top right panel: Green, bottom left panel: Red, and bottom right panel: NIR.

Using those best linear functions and the valid values in the source data channel, it is possible to generate estimates of the missing and poor values in the target data channel. Figures 12 and 13 exhibit the RGB composite map of the first three spectral bands and the B&W map of the NIR spectral band of the CA camera, before and after the replacement of missing values.

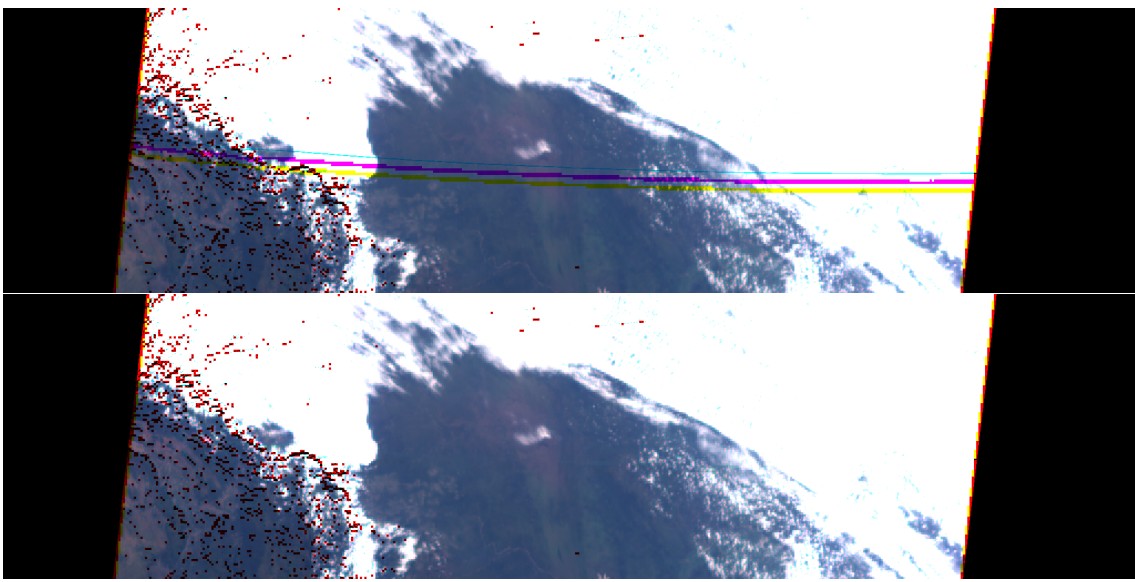

**Figure 12.** RGB maps of the 3 visible data channels for GM-P168-O065487-B110-CA before (top) and after (bottom) replacing the missing and poor data values by the best estimates computed as described above.

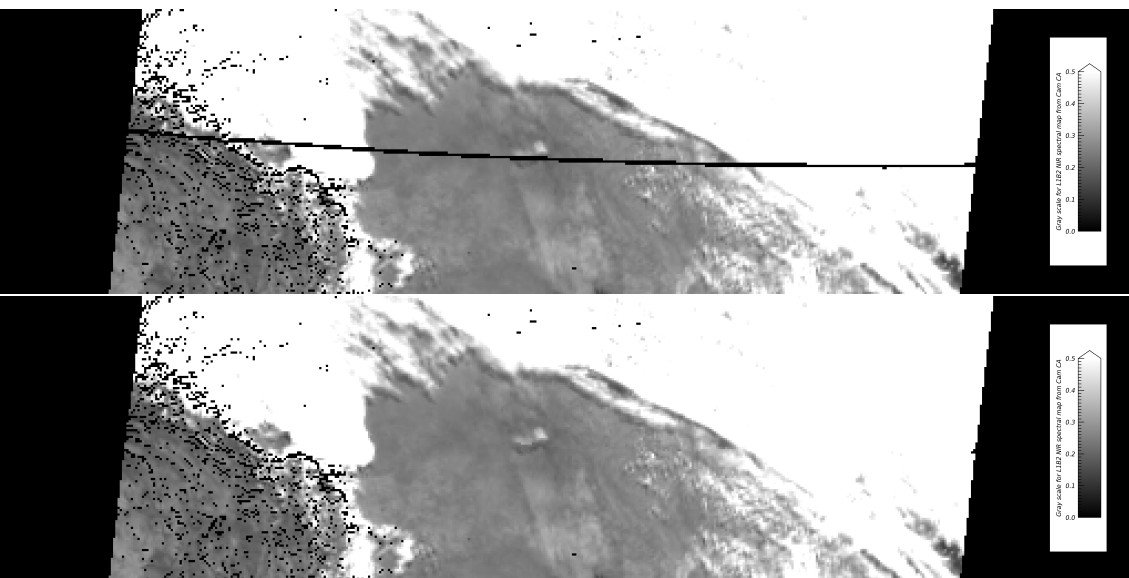

**Figure 13.** Maps of the GM-P168-O065487-B110-CA-NIR Brf data channel before (top) and after (bottom) replacing the missing and poor data values by the best estimates computed as described above.

 ## 4.3 Global Mode, Path 168, Orbit 2111, Block 110, Camera BA

The next instance exhibits the most severe case of missing values encountered so far in our investigations. Figure 14 shows the RDQI values for the four spectral bands of the BA camera for P168, O002111 and B110.

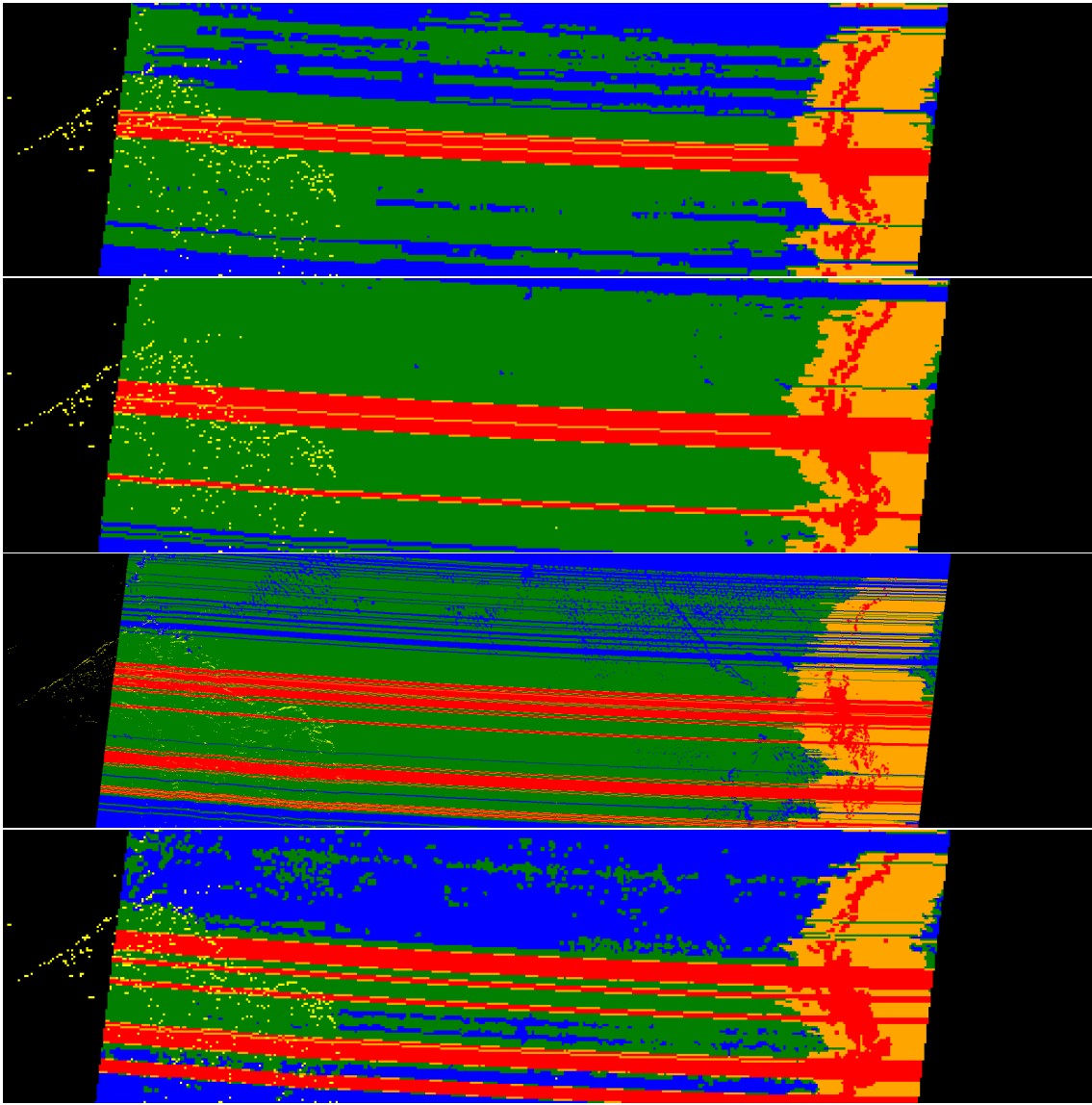

**Figure 14.** Maps of original RDQI values for GM, P168, O002111, B110 and Camera BA, in the Blue, Green, Red and NIR spectral bands, from top to bottom. Linear dimensions are the same as in Figure 1, and color coding is as described in the legend of Figure 10.

The original bidirectional reflectance factor product for these same four spectral bands are shown in Figure 15: it can be seen that missing values severely affect the usability of the data acquired by this camera.

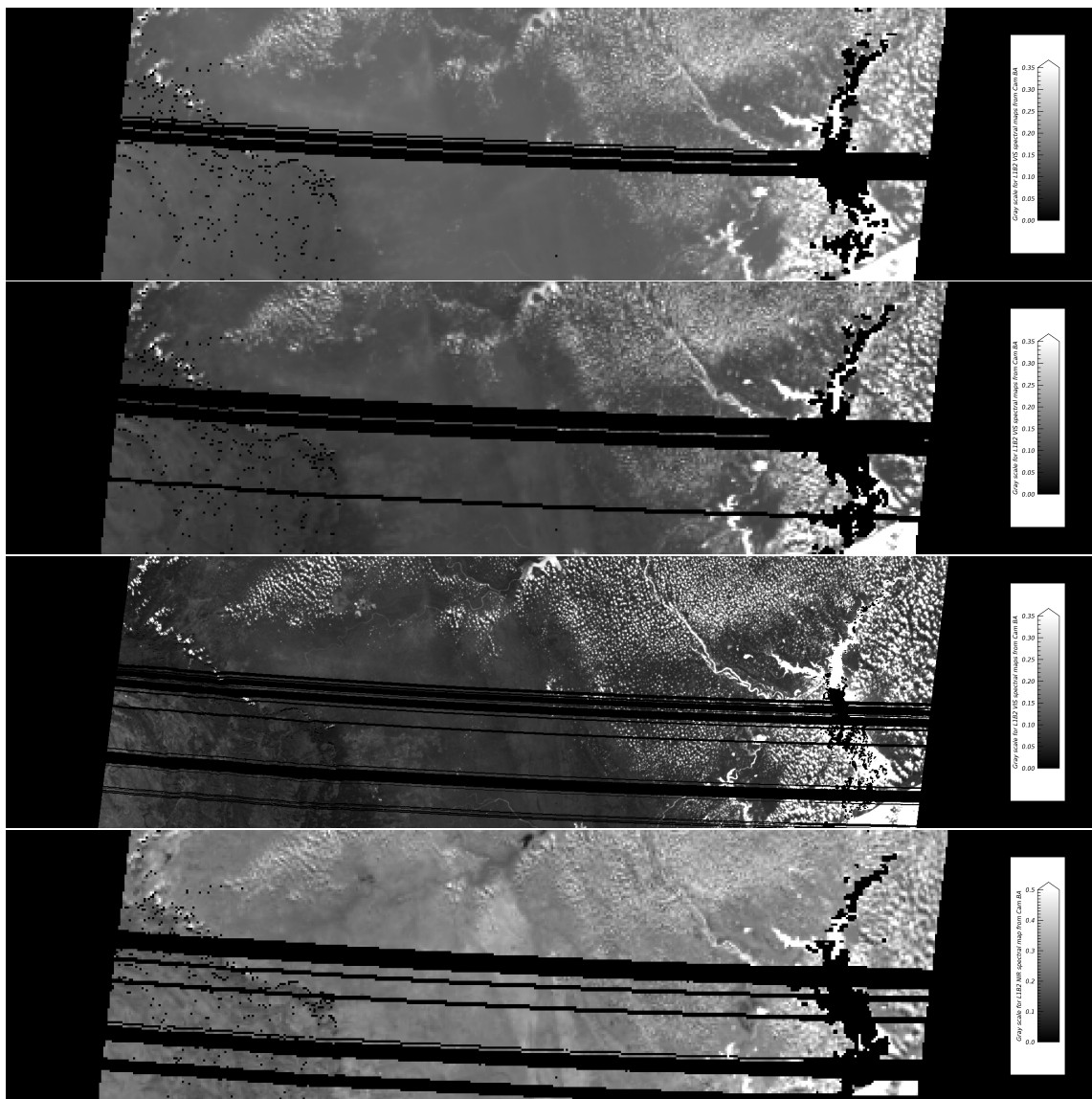

**Figure 15.** Maps of the original ToA bidirectional reflectance factor values for GM, P168, O002111, B110 and Camera BA, in the Blue, Green, Red and NIR spectral bands, from top to bottom. Linear dimensions are the same as in Figure 1, and gray scale stretching is the same as before: it assigns black to a null reflectance and white to reflectances of 0.35 or more in the blue, green and red spectral bands, or 0.50 in the near-infrared spectral band.

Figure 16 exhibits the updated bidirectional reflectance factor values in each of the four spectral channels after processing those data as described above (exceptionally allowing for 8 successive attempts due to the significant overlap between the areas affected by missing values).

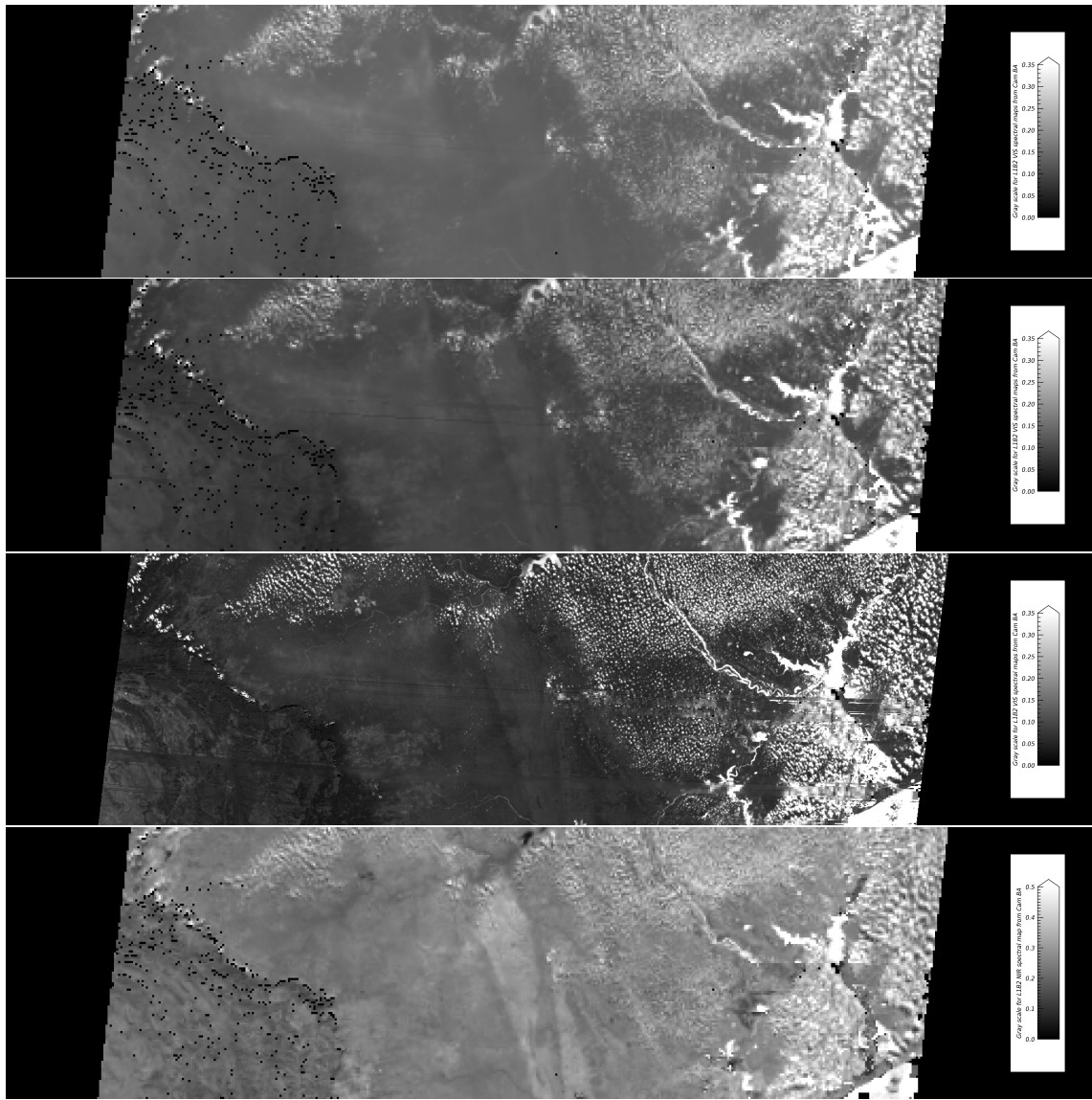

**Figure 16.** Maps of the updated ToA bidirectional reflectance factor values for GM, P168, O002111, B110 and Camera BA, in the Blue, Green, Red and NIR spectral bands, from top to bottom. Achieving this level of replacement required 8 iterations (see text for details). Linear dimensions are the same as in Figure 1, and gray scale stretching is the same as before: it assigns black to a null reflectance and white to reflectances of 0.35 or more in the blue, green and red spectral bands, or 0.50 in the near-infrared spectral band.

## 5 Algorithm evaluation

While the results exhibited in the previous section may be visually impressive, a key remaining question is to assess the
390 numerical accuracy of this procedure. Since it is not possible to evaluate the pertinence of the replaced values when the original

data are missing, the process will be appraised by artificially introducing missing values in files that do not have missing lines in the first place, and then comparing the reconstructed values against the original ones.

The original valid data are first recorded separately, and missing values are inserted in MISR GLOBAL MODE data files for PATH 168, ORBIT 68050 and BLOCK 110 between lines 30 and 34 of the Green spectral band in Camera CF, lines 100 to 110 of the Red spectral band in Camera AN, and lines 50 to 54 of the NIR spectral band in Camera DA. Figure 17 shows the corresponding maps, with black missing lines clearly visible. Note the difference between those straight artificial missing and the curved lines that occur when data are missing following an instrument issue, as well as the rather larger areas affected.

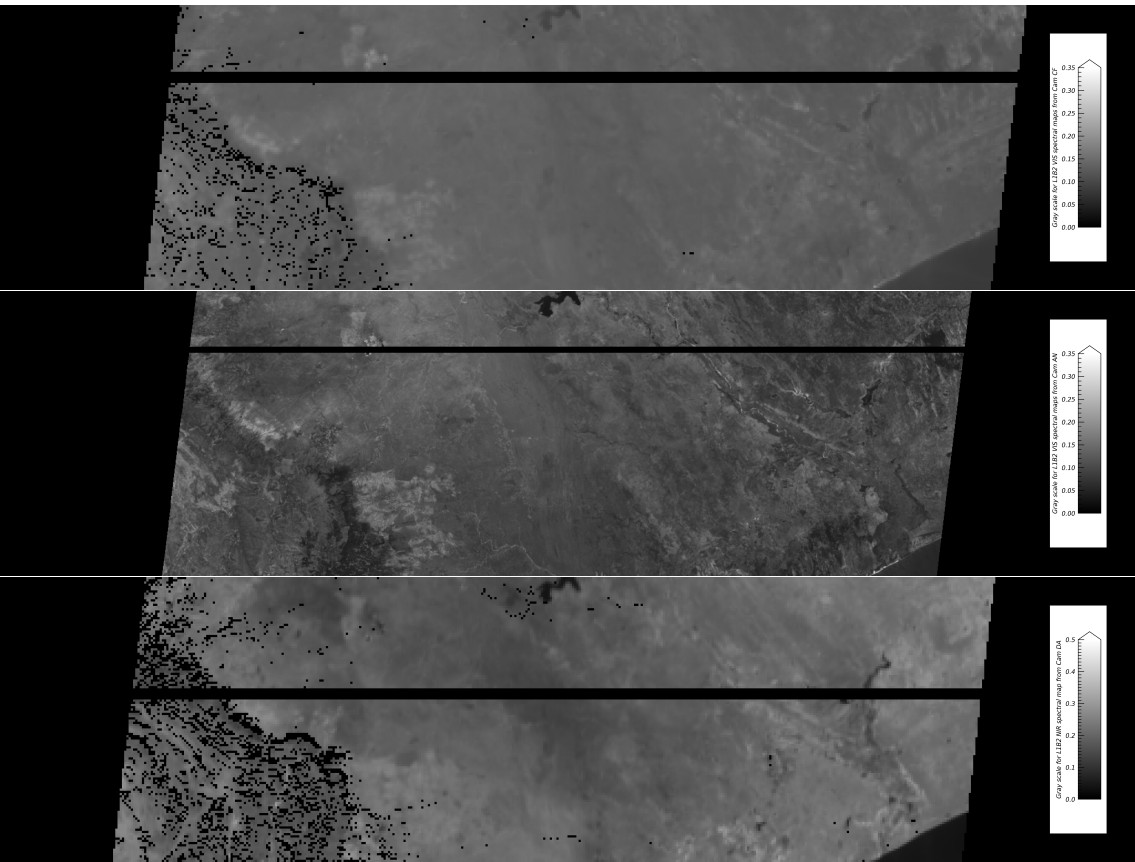

**Figure 17.** Maps of the MISR L1B2 data channels for the CF/Green, AN/Red and DA/NIR data channels of P168, O068050 and B110, where missing data have been inserted artificially for the purpose of testing the algorithm described in this paper. Linear dimensions are the same as in Figure 1, and gray scale stretching is the same as before: it assigns black to a null reflectance and white to reflectances of 0.35 or more in the blue, green and red spectral bands, or 0.50 in the near-infrared spectral band.

These data files are then processed as before, and the three left panels of Figure 18 exhibit the correlation statistics for the data channels that are best correlated with those that need to be updated. The three panels shown on the right compare the original values in abscissa to the reconstructed values in ordinates.

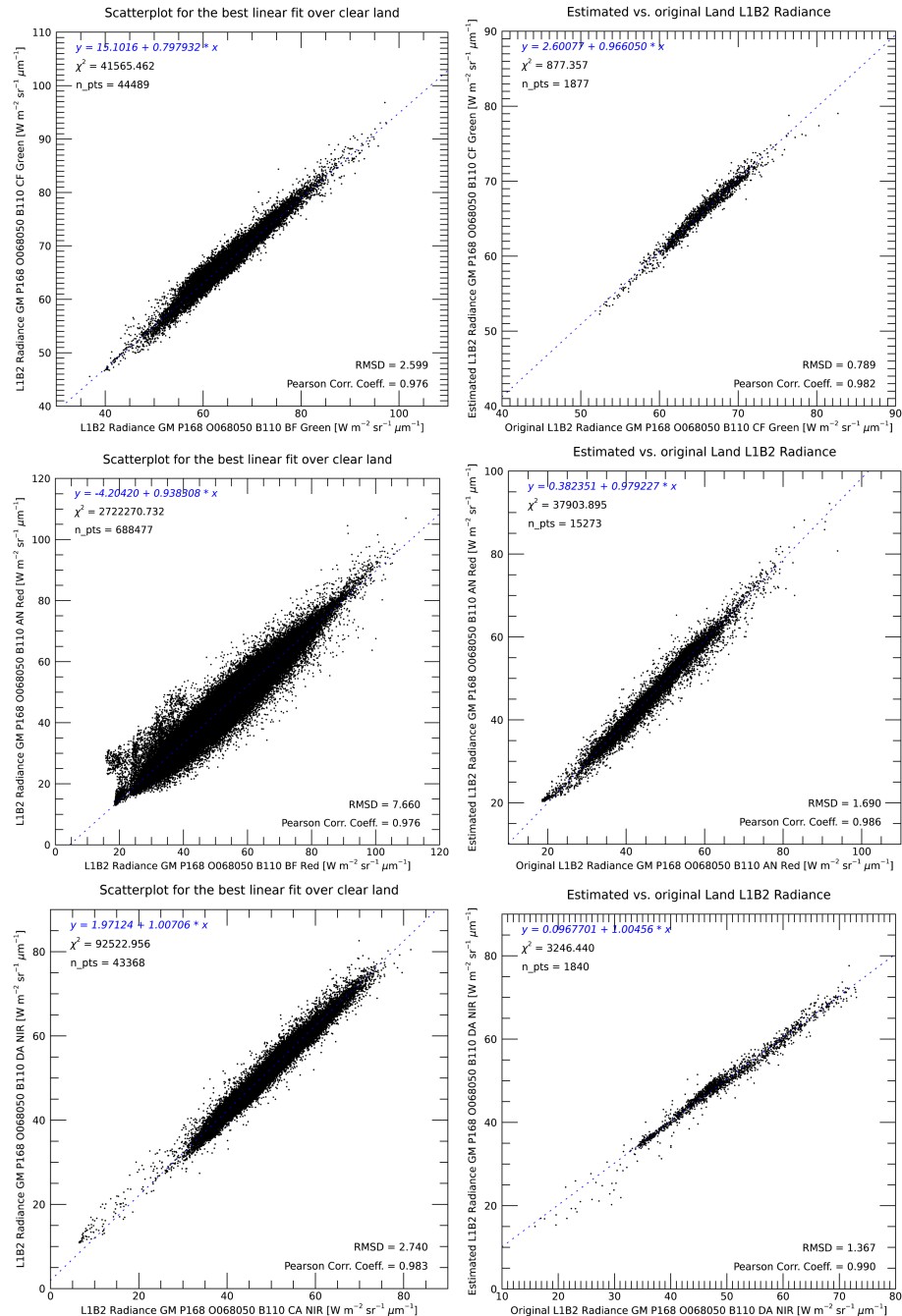

**Figure 18.** Scatterplots of the reconstructed versus the original values in the 3 data channels where missing values were artificially introduced as shown in Figure 17: the top, middle and bottom panels concern the CF/Green, AN/Red and DA/NIR data channels, and in each case, the left panel shows the best predictor (abscissa) for the affected data channel (ordinate), while the right panel exhibits the relation between the original (abscissa) and the reconstructed (ordinate) values, in the areas where missing values were artificially introduced. Note that the ranges of values are different for each plot in order to optimize the view of the data.

Figure 19 exhibits the maps of those three data channels, where the missing data have been replaced by best estimates obtained as described above.

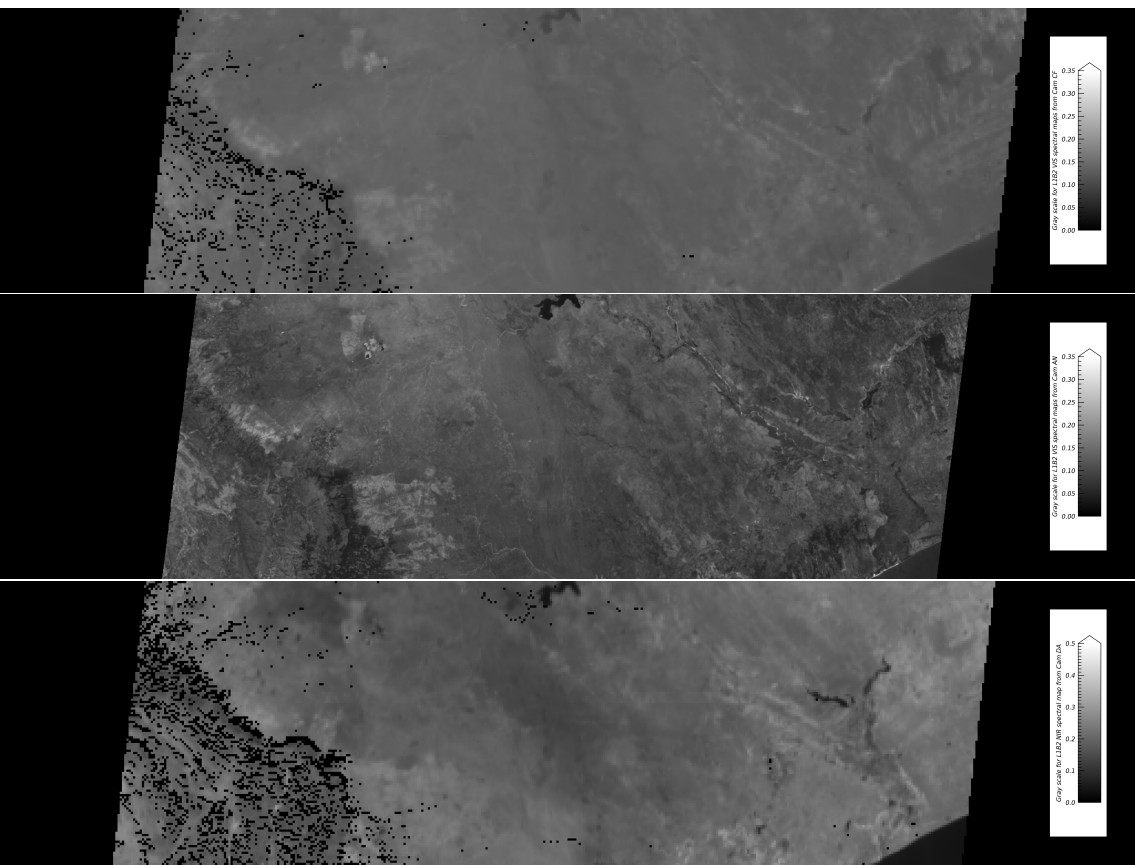

**Figure 19.** Maps of the MISR L1B2 data channels for the CF/Green, AN/Red and DA/NIR data channels of P168, O068050 and B110, where artificially inserted missing data have been replaced by best estimates, using the algorithm described in this paper. Linear dimensions are the same as in Figure 1, and gray scale stretching is the same as before: it assigns black to a null reflectance and white to reflectances of 0.35 or more in the blue, green and red spectral bands, or 0.50 in the near-infrared spectral band.

Similar results have been obtained in all other cases inspected. A second example is considered next, for the case of P168, O065487, B110 documented in Figures 10, 11, 12 and 13, which includes significant cloud cover. In that case too, the original
data channels CF/Green, AN/Red and DA/NIR do not suffer from any missing values, so the same patterns of missing data as described above were artificially inserted in the original data, which were then processed as indicated above. Table 5 provides the number of missing points effectively replaced over clear land masses, the Root Mean Square Difference and the Pearson correlation coefficient between the original and the reconstructed data, as well as the $\chi^2$, the unreduced chi-square statistic that expresses the goodness-of-fit of the linear relation between those to data sets. As noted before, the $\chi^2$ statistics for the AN/Red
data channel is higher due to the much larger number of pixel values involved. Note also that the number of clear land values

effectively replaced depends on the cloudiness present in the scene and therefore on the position of the artificially introduced lines of missing data with in the BLOCK.

**Table 5.** Performance statistics showing the high correlation between the original and the reconstructed MISR L1B2 radiance data over clear land masses, when missing data are artificially introduced in the original data files, for GLOBAL MODE P168, O065487, B110. The numbers of clear land values involved depend on the cloudiness present in the scene, and therefore on the position of the artificially introduced lines of missing data with in the BLOCK.

| Channel | # points | RMSD | PCC | $\chi^2$ |
|---------|----------|-------|-------|------|
| CF/Green | 165 | 3.915 | 0.990 | 574 |
| AN/Red | 1136 | 2.415 | 0.990 | 2762 |
| DA/NIR | 600 | 2.632 | 0.930 | 3512 |

## 6 Conclusions

MISR's L1B2 Georectified Radiance Product (GRP) ellipsoid- or terrain-projected GLOBAL and LOCAL MODE files occasionally include missing or poor data values, in one or more cameras, most often in several or all spectral bands. Their occurrence may result from multiple causes, but the most frequent is the switching between GLOBAL and LOCAL MODE. When a particular area is methodically acquired in LOCAL MODE, as is the case for the Skukuza site in the Kruger National Park, South Africa, those BLOCKs are systematically contaminated by such missing data.

Every radiance measurement acquired by the MISR instrument is scaled into a 14-bit unsigned integer, to which a 2-bit Radiometric Data Quality Indicator (RDQI) is then appended. Missing data are assigned a specific 16-bit scaled radiance code of 65523, equivalent to a 14-bit code of 16380 with an RDQI of 3 attached, while measurements of dubious value (e.g., possibly affected by sun glint) are identified with an RDQI of 2.

The process that generates these missing data points generally affects most or all of the four spectral bands but for only one or two (rarely three) of the nine cameras. However, due to the specific design of the instrument's focal planes, these missing lines affect different geographical locations in different spectral bands and cameras. Consequently, while a particular measurement may be missing in a given camera and spectral band, measurements may be available in other spectral bands or cameras for the same location at essentially the same time. The high correlation between some of the data channels then permits the estimation of those missing values.

A set of software functions, written in the IDL[TM] language[1], was developed to implement the algorithms described above and process MISR L1B2 GRP data to replace those missing values. A selection of cases was exhibited to visually show that the replacement process works well, and a series of tests, based on the artificial insertion of missing data (in files that do not contain any) permitted to show quantitatively the performance and accuracy of the procedure.

---

[1]IDL is a Trademark of Harris Geospatial Solutions, Inc.

As an aside, radiance measurements rated as 'poor', i.e., with an RDQI of 2, have often been found to cluster along those missing lines. These values could also be optionally be replaced by estimates, following the same procedure described above for missing data. The software allows this too, though in that case there is no obvious process for evaluating the results, i.e., there is no objective benchmark to decide whether the original value or the one suggested by the algorithm is closer to the nominal "true" value. The software does allow for the direct comparison between the original and the reconstructed values, through statistics and scatterplots. Users can experiment with both approaches and evaluate whether one or the other yields better or more appropriate results for their applications.

If missing or poor values are effectively replaced by estimates in the updated L1B2 data set, the RDQI assigned to the corrected scaled radiances in L1B2 data buffers is set to 1.

The proposed method to replace missing and optionally poor data in MISR GLOBAL or LOCAL MODE files is generally applicable, and has been implemented in the MISR-HR processing system. The materials in this paper are applicable to Version 2.2.0 of the software, unless noted explicitly in the in-line documentation of the IDL functions. All MISR L1B2 input files are available from NASA's LaRC ASDC Distributed Active Archive Center (DAAC), and routines to explore and address those issues are published on the first author's GitHub web page at https://github.com/mmverstraete.

*Code availability.* This paper describes Version 2.2.0 of the L1B2 IDL functions available in the following GitHub repository: https://github.com/mmverstraete/MISR_L1B2 (last access: 12 March 2020); DOI: https://doi.org/10.5281/zenodo.3519988 (Verstraete (2019b))

These functions, in turn, depend on more basic functions contained in separate repositories, listed here in order of increasing complexity:

- https://github.com/mmverstraete/Macros (last access: 12 March 2020);
  DOI: https://doi.org/10.5281/zenodo.3239995 (Verstraete (2019e))

- https://github.com/mmverstraete/Utilities (last access: 12 March 2020);
  DOI: https://doi.org/10.5281/zenodo.3239999 (Verstraete (2019f))

- https://github.com/mmverstraete/MISR_Tools (last access: 12 March 2020);
  DOI: https://doi.org/10.5281/zenodo.3240005 (Verstraete (2019d))

- https://github.com/mmverstraete/MISR_AGP (last access: 12 March 2020);
  DOI: https://doi.org/10.5281/zenodo.3519897 (Verstraete (2019a))

- https://github.com/mmverstraete/MISR_RCCM (last access: 12 March 2020);
  DOI: https://doi.org/10.5281/zenodo.3240017 (Verstraete (2019c))

Each function contained in those repositories contains abundant in-line documentation to describe every processing step. An HTML file documenting each function in a standardized format is also available in each repository. Functions in the more elaborate repositories may depend on routines defined in more basic repositories. It is thus recommended to download and install the contents of all those repositories to have access to the full set of tools. Please remember also that those tools are under active development and may be updated in time.

The software code to update the RCCM data product further depends on additional external software resources, as documented in Section 6 (Code availability) of Verstraete et al. (2020).

Lastly, for the benefit of users who may not have access to an IDL license, a license-free, self-contained, stand-alone, executable version of the software described above, using the IDL Virtual Machine technology, is available from the research data repository of GFZ Data Services (see Verstraete and Vogt, 2020; https://doi.org/10.5880/fidgeo.2020.011). A user manual describing the acquisition, installation and use of this software is also available from the same source.

*Data availability.* All MISR data, including the RCCM and the L1B2 radiance data products mentioned in this paper, were obtained from the Atmospheric Science Data Center in the NASA Langley Research Center. These data sets are openly available from the ASDC website at https://eosweb.larc.nasa.gov/project/misr/misr_table (last access: 12 March 2020). The references for those data sets are Diner et al. (1999b) (https://doi.org/10.5067/Terra/MISR/MIRCCM_L2.004) for RCCM and Jovanovic et al. (1999) (https://doi.org/10.5067/Terra/MISR/MI1B2T_L1.003) for L1B2.

Two sets of resources have been made available on the research data repository of GFZ Data Services in conjunction with this paper. (A), the first set (Verstraete et al., 2020; https://doi.org/10.5880/fidgeo.2020.012), includes five items: (A1), a compressed archive, `L1B2_Out.zip`, containing all intermediary, final and ancillary outputs created while generating the figures of this paper; (A2), a user manual, `L1B2_Out.pdf`, describing how to install, uncompress and explore those files; (A3), a compressed archive, `L1B2_Suppl.zip`, containing a similar set of additional results, for eight other sites, spanning a much wider range of geographical, climatic and ecological

conditions, to show that the proposed algorithm works as well over areas other than Southeastern Africa; (A4), a companion user manual, `L1B2_Suppl.pdf`, describing how to install, uncompress and explore those additional files; and (A5), a separate input MISR data archive, `L1B2_input_68050.zip`, for PATH 168, ORBIT 68050. This latter archive is usable with (B), the second set (Verstraete and Vogt, 2020; https://doi.org/10.5880/fidgeo.2020.011), which includes (B1), a stand-alone, self-contained, executable version of the L1B2 correction codes, `L1B2_Soft_Win.zip`, using the IDL Virtual Machine technology that does not require a paid IDL license, as well as

(B2), a user manual, `L1B2_Soft_Win.pdf`, to explain how to install, uncompress and use this software.

*Author contributions.* MMV codeveloped the algorithm, ensured software testing, and edited the initial and final draft of the manuscript; LAH was responsible for the codevelopment of the algorithm and updating the manuscript; VMJ contributed the detailed information on the origin of the missing data and the interpretation of the RDQI, and commented on earlier drafts of the manuscript.

*Competing interests.* The authors declare no conflict of interest.

*Disclaimer.* The IDL source code software mentioned above (obtainable from the GitHub web site) is made available under the MIT license, which states, in part, that "The software is provided *as is*, without warranty of any kind, express or implied, including but not limited to the warranties of merchantability, fitness for a particular purpose and noninfringement. In no event shall the authors or copyright holders be liable for any claim, damages or other liability, whether in an action of contract, tort or otherwise, arising from, out of or in connection with

the software or the use or other dealings in the software." Please read carefully and abide by the full license before downloading and using

this software.

*Acknowledgements.*   The first two authors (MMV and LAH) are greatly indebted to Bob Scholes (Global Change Institute (GCI) at the University of the Witwatersrand) for his unconditional scientific support for the MISR-HR project over the past decade and to Barend Erasmus, director of GCI, for sponsoring yearly visits to South Africa during the period 2016–2019. Peter Vogt, from the European Commission Joint Research Centre (JRC), was instrumental in generating a self-contained, stand-alone, executable version of the L1B2 correction software

based on the IDL Virtual Machine technology. Hugo De Lemos, from the GCI, tested the IDL Virtual Machine package. Last but not least, the first author (MMV) thanks David J. Diner of JPL, principal investigator of the MISR instrument, for co-opting him into the MISR Science Team some 25 years ago and for his encouragements ever since.

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
