# Peer review of "Improving the usability of the Multi-angle Imaging SpectroRadiometer (MISR) L1B2 Georectified Radiance Product (2000–present) in land surface applications"

_Earth System Science Data, 2019_

## Referee Comment (RC1) · Anonymous Referee #1 · 1 Mar 2020

Review of Verstraete et al., Improving the usability of the MISR L1B2 Georectified Radiance Product

This paper characterizes and proposes a way to address a significant issue with the 20+ year MISR radiance data record. It is a useful contribution to the literature and is appropriate for ESSD. Some clarification would be helpful, as a great deal of MISR-specific jargon that appears in the text is not explained. My suggestions are mostly for these clarifications, though there are a few more substantive suggestions included in the specific notes. One in particular relates to characterizing the results for more than a single region, and perhaps using a neural net approach to address regions where

the surface is more heterogeneous on 10-100 km spatial scales. A great deal of effort has gone into identifying the issue and developing the algorithm, as presented in this paper, and it will be so much more helpful if the assessment, and possibly also the applicability of the algorithm itself, were more general.

Specific notes:

P2, Line 33-35. You might be more specific here, just for completeness, something like: "In its default Global Mode, for which the instrument acquires data continuously on the day side, all four spectral bands in the nadir camera, and the red spectral bands in the eight off-nadir cameras, are downloaded from the spacecraft at the native resolution of the instrument (275 m). The other 24 channels are averaged on board and downloaded at 1.1 km resolution to reduce the overall data rate." I now see that you get to this in Section 2.1; it might be better to deal with the spatial resolution once, rather than having a vague statement in the introduction and a more specific description one page later.

P3, line 59. You might reference the appropriate ATBD here.

P3, line 80. I know what it is, but you need to define theta_s in Equation 1.

P3, lines 84-87. This statement is also repeated, in this case almost verbatim, from the introduction. Might be better to have it in one place or the other.

P4, Figure 1 caption. You might explain what the "DF" camera is. Might as well also indicate which way is north, given that the caption refers to the "western side" of the image.

P5, after Table 1. You might also explain the criteria for RDQI values of 1 and 2, as these values are considered in section 2.3.

P6, line 115. In which context?

P6, line 118. You might explain what "Path" means in this context.

P8, line 156. You might explain what the MISR-HR product is. I realize a reference is

given, but a sentence or two here would probably be helpful to the reader.

P8, line 161. You might explain what the "AF" and "CA" cameras are.

P8, line 164. Might be: "...what the original measurements would likely have been."

P11, line 236 ff. This seems like an ideal problem to address with a neural net approach, i.e., identifying geophysically similar source pixels in a region upon which to build a correlation function to fill in missing channels in the target pixels. This would avoid the limitations already described in the paper of using the static AGP and illustrated in Figures 6-8. The remaining scatter could be reduced further, as "clear land" alone might offer fairly crude characterization of the surface reflectance spectral-angular dependence in heterogeneous land scenes, especially where multiple ecosystems are present.

P15, line 266. Land areas are apparently of primary interest to the authors (actually, one specific land area, Path 168 Block 110, based on the examples presented), but as you have gone through the trouble of generating an algorithm that seems generic to surface type, why not do the complete problem and at least show the results for other surface types? Further to this point, the examples presented in the paper are all for a single region. If the algorithm is to have more general applicability, it needs to be assessed in a least a few different types of regions, even over land, such as other Blocks where the surface is covered by multiple, distinct ecosystems. (See the note on P11, line 236 above.)

P16, line 289. What are the criteria for a source value to be considered "valid"? I'm a little confused here as to whether a single source pixel is used to fill the missing target values, or whether a statistical summary of multiple source pixels is used. I assumed from the text (e.g., Table 3) it is the latter, but the wording here seems ambiguous. And line 306 seems to indicate a maximum of just four source pixels are used.

P16, line 209. I'd suggest that the filled target channels be given a different RDQI

value, perhaps 4, to indicate that the value has been estimated from other pixels rather than directly measured by MISR.

P25, Figure 18. If I understand correctly, the left plots show the relationship between the predictor and the original target pixel values (before they were removed), and the right panels show the relationship between the original and the replacement target values. It would be easier to interpret these plots if the original were plotted on the same axis in all plots, rather than on the vertical axis for the left plots and the horizontal axis for the right plots. Also, I'm unclear how the replacement is closer to the original than the predictor, upon which the replacement is presumably based. Perhaps I simply do not understand what is plotted here.

---

## Referee Comment (RC2) · Anonymous Referee #2 · 15 Mar 2020

This is a helpful paper for "power users" of MISR Global Mode products who are able to work with L1B2 products as an input to subsequent processing to higher-level products.

It is not clear if this algorithm is going to be incorporated into the operational MISR level-1 re-processing system which is due to take place at some point in 2020.

It is also not clear how much of the MISR data is affected as a function of time as there are no statistics on the number of Local Mode sites, other than the surprising statement that there is only one LM site per orbit with 14 orbits per day.

There is also no assessment of what impact a Local Mode site would have depending on where in the MISR Block it is located. For example, if this LM site were located near

one of the boundaries of a MISR Block, would the method still work?

Finally, have the authors considered that the reason why the RED channel shows the largest spread is because there are 4 times as many pixels for the Red than the other 3 channels.

Annotation summary of uploaded file: — Page 2 —

Caret, Anonymous: sea ice

Note (yellow), Anonymous: Is it possible to provide an estimate of what percentage of data is local mode?

Strikeout (red), Anonymous: the

Caret, Anonymous: a

Strikeout (red), Anonymous: atmospheric and

Note (yellow), Anonymous: atmospheric products are revived from ellipsoid-correct not terrain-corrected products

Strikeout (red), Anonymous: e

Caret, Anonymous: is product

Strikeout (red), Anonymous: latter

Note (yellow), Anonymous: Why is there no outline of what the following sections discuss?

— Page 3 —

Caret, Anonymous: nd corrected for terrain relief effects using

Strikeout (red), Anonymous: on

Note (yellow), Anonymous: Are the authors sure about this?

Note (yellow), Anonymous: Is this the same as a so-called block? If so, say so.

Caret, Anonymous: or over urban areas of pollution

Highlight (color #FF4FE8), Anonymous: swath. Repl

— Page 7 —

Note (yellow), Anonymous: Given that this is predictable why is a tag not added to the data for these?

— Page 8 —

Note (yellow), Anonymous: But why is the cloud cover at 17.6km? I thought that RCCM and ASCM was at 275m?

Note (yellow), Anonymous: Please indicate the location and also explain why most of the area is black?

— Page 9 —

Note (yellow), Anonymous: I thought that NASA removed this after the end of February 2020?

— Page 10 —

Caret, Anonymous: and over clouds

Caret, Anonymous: other

Strikeout (red), Anonymous: others

— Page 11 —

Caret, Anonymous: to

— Page 12 —

Note (yellow), Anonymous: These would be much more useful if they employed a

colour-density lookup table to show the relative importance of the off-achromatic pixels.

— Page 14 —

Note (yellow), Anonymous: How is this software operational? Does it run at the DAAC?

— Page 15 —

Caret, Anonymous: ,

Strikeout (red), Anonymous: Brf

Caret, Anonymous: BRF

— Page 16 —

Note (yellow), Anonymous: It would be helpful to add some performance figures here. Also, will this be taken up by NASA in the planned level-1 re-processing?

— Page 19 —

Caret, Anonymous: than

— Page 24 —

Note (yellow), Anonymous: Why not use MODIS BRFs from MOD09 or Landsat-7 taken at the same time as MISR??

— Page 26 —

Strikeout (red), Anonymous: a

(report generated by GoodReader)

Please also note the supplement to this comment:
https://www.earth-syst-sci-data-discuss.net/essd-2019-210/essd-2019-210-RC2-supplement.pdf

[Figure]

**Supplement:**

[revised manuscript text omitted]

---

## Author Comment (AC1) · 23 Mar 2020

**Responses to Reviewers #1 and #2**

Michel M. Verstraete[1], Linda A. Hunt[2], and Veljko M. Jovanovic[3]

[1]Global Change Institute (GCI), University of the Witwatersrand, Braamfontein, Republic of South Africa.
[2]Science Systems and Applications, Inc. (SSAI), Hampton, VA 23666-5845, USA.
[3]NASA Jet Propulsion Laboratory (JPL), Pasadena, CA 91109, USA.

**Correspondence:** Michel M. Verstraete (Michel.Verstraete@wits.ac.za or MMVerstraete@gmail.com)

Dear Reviewers,

Thanks a lot for your very useful comments on the manuscript entitled 'Improving the usability of the MISR L1B2 Geo-rectified Radiance Product (2000–present) in land surface applications' (essd-2019-210). We have addressed your suggestions as explained below, and various Sections have been substantially reworked. Additional results for different environments have

5 also been made available through the GFZ Data Services web portal.

**1 Reviewer #1**

**1.1 Generic comments**

Two generic requests are initially expressed: The first one is to show that the solution proposed in the manuscript to replace missing values in MISR L1B2 GRP data files is applicable to areas and environments other than those explored in Southeastern

10 Africa, as all examples and Figures in the manuscript concerned that region. To address this point, we have compiled an additional archive of four cases, spread in latitude between Iraq and Namibia and spanning a variety of surface types (more on this below). This additional archive is provided through the GFZ Data Services portal at http://doi.org/10.5880/fidgeo.2020.012, which already hosted all the intermediary, final and ancillary results generated in the course of producing the Figures of this paper. The second request is related to 'perhaps using a neural net approach to address regions where the surface is more

15 heterogeneous on 10-100 km spatial scales'. Both points are discussed in greater detail below, as they arose again in specific remarks.

**1.2 Specific remarks**

The manuscript itself has been edited as follows (page and line numbers refer to the original version and correspond to your comments):

20 - p. 2, lines 33–35: *You might be more specific here, just for completeness, something like: "In its default Global Mode, for which the instrument acquires data continuously on the day side, all four spectral bands in the nadir camera, and the red spectral bands in the eight off-nadir cameras, are downloaded from the spacecraft at the native resolution of the instrument (275 m). The other 24 channels are averaged on board and downloaded at 1.1 km resolution to reduce the*

*overall data rate." I now see that you get to this in Section 2.1; it might be better to deal with the spatial resolution once,*
*rather than having a vague statement in the introduction and a more specific description one page later.*

Response: There was indeed some degree of repetition between the introduction and the materials in Subsection 2.1 (lines 59–68). It has proven cumbersome to move the full explanation of the Global and Local Modes, together with the spatial resolution differences, to the introductory section without deforcing the latter section. Instead, we have opted to simplify the introduction and just hint at the problem of a variable number of missing data to be explored further in the immediately following section. The duplication is removed and the core of the argument thus remains in Subsection 2.1.

- p. 3, line 59: *You might reference the appropriate ATBD here.*

  Response: We have added two references here: the ATBD for specialists, as requested, and the Data Products Specifications document, which is the primary source of information on MISR data products for the majority of users.

- p. 3, line 80: *I know what it is, but you need to define $\theta_s$ in Equation 1.*

  Response: We have added the definition of the solar zenith angle in the text following Equation 1.

- p. 3, lines 84–87: *This statement is also repeated, in this case almost verbatim, from the introduction. Might be better to have it in one place or the other.*

  Response: Here too, we have opted to simplify the introductory materials (Section 1) and to keep the full description of the process involved in Section 2.

- p. 4, caption of Figure 1: *You might explain what the "DF" camera is. Might as well also indicate which way is north, given that the caption refers to the "western side" of the image.*

  Response: We have addressed this suggestion as follows:

  ○ The 9 cameras of MISR are now explicitly named in the introduction (Section 1), where additional details on the specific camera angles and spectral band positions have also been added for completeness.

  ○ A sentence has been added to the caption of Figure 1 to indicate that North is approximately pointing 'upward'.

  ○ Another sentence has also been inserted in the text discussing Figure 1 (page 5) to refer to Figure 2 of the companion paper Verstraete et al. (2020) on the RCCM data product, which shows where this particular Block is located in Southern Africa.

- p. 5, after Table 1: *You might also explain the criteria for RDQI values of 1 and 2, as these values are considered in section 2.3.*

  Response: RDQI values are derived from a rather complex analysis of raw data, which is described in detail in the MISR Level 1 ATBD. Including a full discussion of that mechanism is well beyond the scope of this paper, but we have added a citation and a summary sentence which represents our practical experience with those values, namely that a RDQI value

of 1 refers to either a minor increase in the uncertainty of the radiance value or a possible slight mis-registration of the pixel, while a value of 2 indicates a larger radiance uncertainty, for instance in the presence of sun glint reflected from a still water surface.

- p. 6, line 115: *In which context?*

    Response: The expression 'in this context' refers to this particular paper: the text has been edited to clarify this point.

- p. 6, line 118: *You might explain what "Path" means in this context.*

    Response: We have also added a sentence near the start of the Section 1 to briefly define the concept of Path for the Terra platform.

- p. 8, line 156: *You might explain what the MISR-HR product is. I realize a reference is given, but a sentence or two here would probably be helpful to the reader.*

    Response: We have added a sentence to outline the main purpose of the MISR-HR processing system.

- p. 8, line 161: *You might explain what the "AF" and "CA" cameras are.*

    Response: As explained above, all nine cameras of the MISR instrument are now named and characterized in terms of angular settings and spectral bands in the introductory section of the paper.

- p. 8, line 164: *Might be: "... what the original measurements would likely have been."*

    Response: We have edited this sentence as suggested.

- p. 11, line 236–245: *This seems like an ideal problem to address with a neural net approach, i.e., identifying geophysically similar source pixels in a region upon which to build a correlation function to fill in missing channels in the target pixels. This would avoid the limitations already described in the paper of using the static AGP and illustrated in Figures 6-8. The remaining scatter could be reduced further, as "clear land" alone might offer fairly crude characterization of the surface reflectance spectral-angular dependence in heterogeneous land scenes, especially where multiple ecosystems are present.*

    Response: We do not agree with the suggestion to use a more detailed land cover map or—equivalently—to rely on a Neural Network (NN) approach to discriminate between multiple types of land surfaces, for the following reasons:

    - The results generated by the procedure described in the manuscript would then depend on an external data set, or on the particular settings of the Neural Network (NN) algorithm. This immediately raises multiple questions: Which land cover map to choose from (there are many options)? How many NN classes would be appropriate (this is largely an arbitrary choice)? How reliable is(are) this(those) map(s) or the results of a dynamically generated NN? Different assumptions would lead to different results.

○ The more surface types distinguished, the smaller the number of data points in each class. Since the processing is always performed on a Block basis, some, if not most, categories defined in such a land cover map would inevitably contain relatively few data points, which would result in less reliable derived statistics. Using a fine-grained land cover map or the NN approach might perhaps be better justified with a hyperspectral instrument, but looking at the Earth from space with an instrument featuring only 4 spectral bands, it is unlikely that multiple types of vegetation (or soils) could be reliably distinguished within the area covered by a single Block. One could of course develop a new land cover classification that takes advantage of the observed spectral and directional variability in all 36 data channels of the MISR instrument, but that would entail an entirely new investigation, well beyond the scope of this paper. A NN approach would also likely introduce artificial contrasts at Block boundaries.

○ The algorithm calls for the computation of correlation statistics and linear fits between the target (containing missing values) and the 35 other data channels of MISR, in order to rank those possible sources in order of decreasing correlation coefficient. Using a land cover (or a NN-derived) map with 12 classes, for instance, would quadruple the computing time required to select the most appropriate sources and replacement functions, compared to the currently recommended approach of 3 classes (counting clouds as one category). This might be justifiable for one or a few Blocks, but prohibitive for the systematic reprocessing of the MISR archive over a large area.

○ It should also be kept in mind that the MISR L1B2 GRP radiance values analyzed in this context are measurements acquired at the nominal top of the atmosphere: they include the non-negligible contribution from the path radiance of the atmosphere. A NN classification would therefore also be biased by the presence and properties of aerosols and thin cloud veils, which vary substantially in space and time.

○ Last but not least, land cover evolves in time anyway, through processes such as deforestation and desertification, drought and flood, the rapid growth of vegetation in arid zones after a rain event and its quick disappearance due to fire, or changes in crop species and agricultural practices. An approach based on a detailed land cover classification should therefore also include the acquisition and regular updating of the underlying land cover map (or the frequent updating of the NN classification), a challenge in itself. Most importantly, it might then become problematic to compare maps derived for different dates, since they would result from different assumptions about the state of the atmosphere and the implied land cover classification.

- p. 15, line 266: *Land areas are apparently of primary interest to the authors (actually, one specific land area, Path 168 Block 110, based on the examples presented), but as you have gone through the trouble of generating an algorithm that seems generic to surface type, why not do the complete problem and at least show the results for other surface types? Further to this point, the examples presented in the paper are all for a single region. If the algorithm is to have more general applicability, it needs to be assessed in a least a few different types of regions, even over land, such as other Blocks where the surface is covered by multiple, distinct ecosystems. (See the note on P11, line 236 above.)*

Response: The mathematical algorithms and the software codes described in the manuscript are indeed completely generic and oblivious to the geographical location of the scenes, or the time of acquisition of the data. The various

Figures of the manuscript, and the exhaustive results contained in the compressed archive 'L1B2_Out.zip', available from the GFZ data portal at http://doi.org/10.5880/fidgeo.2020.012, already show the applicability of those algorithms to various ecosystems. Nevertheless, to provide further evidence over a broader range of surface types, we compiled another

archive containing additional results for 4 carefully selected cases: Iraq (marsh and arid lands), Kenya (agriculture and tropical forests), South Sudan (grasslands) and Namibia (coastal desert and Atlantic Ocean). Two of them involve largely clear scenes, and the other two include clouds. In the last case, we have also run a test to artificially introduce missing data over deep water and clouds, to demonstrate the performance of the procedure on targets other than continental areas. These results are contained in a new archive 'L1B2_Suppl.zip' (675.5 MB), also available from the same GFZ

portal. Once uncompressed, this new archive expands into 8 subdirectories and takes up 1.8 GB of disk space, providing access to about 2,900 files. We have also drafted a User Manual ('L1B2_Suppl.pdf') for that supplementary archive. This document is also appended to this reply: it explains how to obtain and install this archive, and outlines its contents in some detail.

- p. 16, line 289: *What are the criteria for a source value to be considered "valid"? I'm a little confused here as to whether a single source pixel is used to fill the missing target values, or whether a statistical summary of multiple source pixels is used. I assumed from the text (e.g., Table 3) it is the latter, but the wording here seems ambiguous. And line 306 seems to indicate a maximum of just four source pixels are used.*

Response: The expression 'ensure that this source value is valid' in this context simply means that the source data value is itself not missing or poor (i.e., has an RDQI of 0 or 1). This is now clarified in the paper.

Each of the four lines in Table 3 provides the applicable statistics characterizing the best source data channel for the indicated target data channel. Hence, the best predictor to replace missing values in DF/Green data channel will be the CF/Green data channel, in this case. Some 44,866 pixels with the same geographical coordinates were found to be valid on both data channels of the indicated Block, and the Pearson correlation coefficient is 0.962. The linear fit equation exhibited in the top right panel of Figure 8 is then used to replace missing values in DF/Green. A similar reasoning applies to the other lines in that Table.

Note, however, that each replacement value is based exclusively on a single source value. What may happen is that the preferred source value (i.e., the one from the data channel enjoying the highest correlation coefficient) for a particular geographical location is also missing, for one of the reasons described in Subsection 2.2 of the paper. In that case, the next best data channel is inspected to see if that source pixel at the same geographical location is valid: if so, the statistical results obtained for that source/target couple are used to replace the missing value in the target. This iterative process is repeated until a suitable (RDQI < 2) source value is found, or the upper limit on the number of attempts (typically up to 4 source data channels) allowed to be used, is reached. This latter limit is imposed to avoid cases where the replacement value might be unreliable due to a low correlation coefficient or an excessive dispersion value, as explained in lines 302–309.

150     • p. 16, line 209 (assumed to be line 292): *I'd suggest that the filled target channels be given a different RDQI value, perhaps 4, to indicate that the value has been estimated from other pixels rather than directly measured by MISR.*

     Response: It is not possible to set the RDQI to any other value than 0, 1, 2 or 3 because it is coded as the last 2-bits of the L1B2 scaled radiance value, as described in Table 1 and the surrounding text on p. 5. We have elected to set the RDQI of the replaced values to 1 as a reminder that they are reconstructed rather than originally measured. It is also our

155      experience that relatively few values are assigned an RDQI of 1 by NASA's standard processing system. Hence, most of those values (after being updated as explained in this manuscript), would point to values that were missing and have been replaced by estimates. It is also worth noting that the MISR-HR processing system, which provides the broader context in which these updates are taking place, explicitly keeps track of those replacements in the associated external PIXELFLAGS file. The latter includes a specific bit of information that indicates whether each pixel value is reported as

160      originally available in the MISR data (0) or has been updated to replace a missing value (1). This level of detail is useful for auditing and diagnostic purposes, but would be of little interest for users interested in practical applications.

    • p. 25, Figure 18: *If I understand correctly, the left plots show the relationship between the predictor and the original target pixel values (before they were removed), and the right panels show the relationship between the original and the replacement target values. It would be easier to interpret these plots if the original were plotted on the same axis in all*

165      *plots, rather than on the vertical axis for the left plots and the horizontal axis for the right plots. Also, I'm unclear how the replacement is closer to the original than the predictor, upon which the replacement is presumably based. Perhaps I simply do not understand what is plotted here.*

     Response: This Figure includes three rows of two panels each. The rows refer to the results obtained for each of the three cases of artificially inserted missing values, namely the green spectral band of the CF camera (top row), the red spectral

170      band of the AN camera (middle row), and the NIR spectral band of the DA camera (bottom row).

     In each of those three cases, the left panel shows the relation between the best predictor (source) data channel, shown in abscissa, and the target data channel, shown in ordinates: those results document the strength of the statistical relations used in the replacement process. This is consistent with the custom of graphically representing a function $y = f(x)$ in a 2-D diagram, where the independent variable ($x$, or the source data channel) is laid along the horizontal axis, and the

175      dependent variable ($y$, or the target data channel) along the vertical axis.

     The right panel of each row exhibits the relation between the original values, before they were artificially considered missing, and the estimated replacement values. Here again, the known variable is along the horizontal axis, while the statistically derived one is spread along the vertical axis. Those panels demonstrate that the process of replacement of missing values does in fact yield estimates that are very similar to the actual measurements, in the rather artificial case

180      when one knows the 'true' values. Please note also that none of those axes are directly comparable. For instance, none of the points shown in the right scatterplots have any correspondent in the left scatterplots: they were missing altogether for the purpose of that test, and in fact the values shown along the vertical axis of the right scatterplot are added to the values shown along the vertical axis of the left scatterplot to generate the 'updated' map with few or no missing values.

Lastly, those 3 right panels concern relatively small subsets of points within the target data channel only, specifically those that have been artificially declared missing in the experiment and subsequently replaced by the algorithm, while the left panels involve all non-missing values that share the very same geographical location in the source and the target data channels. These scatterplots are shown side by side to save space, and they encapsulate the gist of the algorithms, but they should not be compared.

**2 Reviewer #2**

**2.1 Generic comments**

The first four generic comments are answered as follows:

- *It is not clear if this algorithm is going to be incorporated into the operational MISR level-1 re-processing system which is due to take place at some point in 2020.*

  Response: This manuscript was originally submitted to the ESSD journal on 27 October 2019, and a request to incorporate these algorithms into the scheduled reprocessing was formally filed at JPL at the same time. This issue was discussed at the annual MISR Science Team meeting in Pasadena on 11 February 2020. It appears that the available budget does not permit the MISR Team to cater for this additional processing within the current funding cycle. However, this task has been identified for funding as part of the proposal for an extension of the Terra/MISR mission until 2026, in the context of the on-going Senior Review. Support from the scientific community to ensure funding for this activity will be appreciated.

- *It is also not clear how much of the MISR data is affected as a function of time as there are no statistics on the number of Local Mode sites, other than the surprising statement that there is only one LM site per orbit with 14 orbits per day.*

  Response: This question has been addressed by significantly expanding the description of Local Mode acquisitions in Subsection 2.1 of the manuscript, which now reads as follows:

  Upon request, the MISR instrument can also be operated in Local Mode (LM). When this mode is activated, each camera is allowed to transmit data in turn at the native spatial resolution of the detectors in all 4 spectral bands, for a limited period of time, corresponding to a scene of about 300 km along-track. LM acquisitions occur mostly over continental areas, and are requested in three broad categories of cases: (1) to systematically assess the performance of the instrument through vicarious calibration exercises over fixed, well documented sites, and throughout the mission, (2) to support field campaigns, over the areas of interest and for the duration of those exercises, or (3) to promote thematic studies, for instance over polluted urban areas, or to monitor a dynamically evolving region. About 100 sites have been identified for LM acquisition over the first 20 years of operation of MISR, with periods of observation ranging from a few weeks to the entire duration of the mission. The number of LM acquisitions per Orbit is a mission design constraint, as each one requires

215 the specific re-programming of the instrument and leads to larger data transfers to the ground segment. In practice, only one LM acquisition is allowed per Orbit on an operational basis. At the time of writing (mid-March 2020), Local Mode acquisitions were undertaken roughly 3 to 8 times a day (out of 14 opportunities). The ASDC maintains a table, updated daily, reporting on the success of LM acquisitions (see https://l0dup05. larc.nasa.gov/public/cgi-bin/DUE/ecs_LocalMode_history_PR.cgi).

220 Further details are provided below, as this question is raised again in a specific remark on page 2 of the manuscript.

- *There is also no assessment of what impact a Local Mode site would have depending on where in the MISR Block it is located. For example, if this LM site were located near one of the boundaries of a MISR Block, would the method still work?*

Response: Neither Global nor Local Mode acquisitions know about or are related to Blocks. The concept of the Block
225 is used to partition the data for storage in a way that accommodates the use of one of the prescribed HDF-EOS data structures, the GRID (as opposed to POINT or SWATH) which are geolocatable. In other words, Blocks are just a logical way to point to subsets of Orbits, they have no bearing on how the observations are acquired in the first place. In fact, inspecting Figure 1 of the manuscript, one can see that this particular LM acquisition does not entirely cover Block 110 of Path 168: a small piece is missing in the Northeastern part of the Block.

230 The algorithm described in this paper is totally independent from the availability of Local Mode data: missing data in Global Mode files are replaced using other data channels acquired simultaneously in the same Global Mode. Local Mode is only discussed in his context because the very process of switching the MISR instrument from Global to Local Mode (and conversely) creates a burden on the computer which causes an order of magnitude more missing data than under a routine acquisition. And, by the way, it is also possible to replace missing data in the Local Mode files: that process will
235 only involve statistical relations between the various data channels of that same LM acquisition.

- *Finally, have the authors considered that the reason why the RED channel shows the largest spread is because there are 4 times as many pixels for the Red than the other 3 channels.*

Response: Yes, of course, the data downloaded from the red spectral channel of the eight off-nadir pointing cameras, as well as from the four spectral channels of the nadir-pointing camera are obtained at the full, native spatial resolution of
240 the instrument, and therefore result in data sets roughly 16 times larger than the other data channels.

**2.2 Specific remarks**

The manuscript itself has been edited as follows (page and line numbers refer to the original version and correspond to your comments):

- p. 2, l. 29: *sea ice*

245 Response: The revised paper now explicitly mentions the cryosphere, including glaciers, ice caps and sea ice, as another scientific field regularly exploiting MISR observations.

- p. 2, l. 35: *Is it possible to provide an estimate of what percentage of data is local mode?*

Response: The Authors are not aware of the existence of published, up-to-date, global statistics about the spatial and temporal distributions of Local Mode acquisitions. One way to get a feel for the amount of LM data compared to GM acquisitions is as follows: If five (out of 14 possible) LM sites are acquired per day, if each acquisition perturbs four (out of 142) Blocks of data, and if 10 (out of 128) lines are missing in those affected Blocks, the proportion of missing data is 0.07% of the total number of observations acquired. The key points to keep in mind are that missing data (1) can occur anywhere and at any time, albeit infrequently, and (2) do occur in large numbers in Blocks located near sites acquired in Local Mode.

The impact of those missing data is more important than the abstract statistics about their prevalence. As has been mentioned before and in the paper, studies encompassing large areas and/or long periods will not be affected because of the plethora of data. However, investigators interested in specific areas and particular periods may be severely affected by missing data if MISR data are being acquired in LM near their site. Figure 3 of the manuscript, as well as similar Figures for the various additional cases presented in the external archive 'L1B2_Suppl.zip', available from GFZ as explained above, document the range of situations to be expected: Block 109 of Path 180, which encompasses the coastal area of Namibia and is not perturbed by LM acquisitions, features 68,759 missing values between February 2000 and late 2019. By contrast, Block 110 of Path 168, which contains the Skukuza flux tower site, includes over 2,143,000 missing values in the same period (even though LM acquisitions only started in 2010).

- p. 2, l. 49: *Why is there no outline of what the following sections discuss?*

Response: An additional small paragraph has been added at the end of the introductory Section to outline the rest of the paper.

- p. 2, l. 51: *Strikeout the*

Response: The word 'the' has been replaced by 'a', though this initial Section has been significantly reworked to address the comments of Reviewer #1.

- p. 2, l. 53: *atmospheric products are revived from ellipsoid-correct not terrain-corrected products*

Response: This statement is incorrect. For instance, the RCCM data product over continental areas is specifically derived using the terrain-projected L1B2 GRP data product: See page 9 of

Diner, David J. and Di Girolamo, Larry and Clothiaux, Eugene E. (1999) 'Level 1 Cloud Detection Algorithm Theoretical Basis', Jet Propulsion Laboratory, California Institute of Technology, JPL D-13397, Revision B, available from https://eospso.gsfc.nasa.gov/atbd-category/45, DOI=10.5067/Terra/MISR/MIRCCM_L2.004.

Ellipsoid-projected radiances are only used over the oceans for this particular purpose. Nevertheless, since this paper is primarily concerned about improving products over continental areas, the text has been modified to focus specifically on that objective.

- p. 3, l. 56: *nd corrected for terrain relief effects using*

Response: This editorial change has been implemented as suggested.

- p. 3, l. 68: *Are the authors sure about this?*

Response: Yes. There are no standard MISR products, at Level 2 or higher, available at a spatial resolution finer than 1100 m. Of course, the MISR-HR processing system, initially described in Verstraete et al. (2012) does deliver a whole range of products at the full native spatial resolution of the instrument (275 m), but those are not standard MISR products: they are custom derived products available for limited areas (currently mostly in South Africa.)

- p. 3, l. 70: *Is this the same as a so-called block? If so, say so.*

Response: As already pointed out above, Local Mode acquisitions do not 'know about' or relate to Blocks, which are logical concepts to deal with large data files, after the observations have been collected. Blocks can be used to designate subsets of GM or LM data files (as demonstrated in Figures 1 and 2). LM acquisitions typically cover an area somewhat larger than two consecutive Blocks, but the northern and southern boundaries of the LM acquisition do not match the pre-defined and fixed Block boundaries.

- p. 3, l. 72: *or over urban areas of pollution*

Response: This change has been implemented as suggested.

- p. 3, l. 81: *Highlighted text: swath. Repl*

Response: The statements on line 80 to 83 of the manuscript were reproduced verbatim from the indicated sources (Bull et al.) The text has been edited to provide the same information in a more readable form, and the opening and closing quotes have been deleted.

- p. 7, l. 142: *Given that this is predictable why is a tag not added to the data for these?*

Response: It is precisely the purpose of the RDQI to inform the user on the quality of the data. Hence, 'poor' data values are associated with an RDQI of 2. The L1B2 GRP data files also include information on the potential risk of contamination by Sun glint, so users wanting to delve into pixel-specific values can analyze that field in conjunction with the RDQI to determine whether the latter rating was assigned because of the former value.

- p. 8, l.1 of the caption of Figure 5: *Please indicate the location and also explain why most of the area is black?*

Response: As indicated in the caption, this map concerns Path 168 and Block 110, i.e., is the very same area already mapped in Figures 1 and 2. The caption of Figure 5 has been updated to make that clear. The text describing Figure 1 of this paper now also includes a pointer to Figure 2 of the companion paper Verstraete et al. (2020), which locates this and a few neighboring Blocks in a broader map of Southern Africa.

All pixels for which there are no data are represented in black (compare again with Figure 1). This includes (1) the individual black dots in the southwestern corner of the map, corresponding to mountain slopes that are not observable by

310  (in this case) one or more cameras, (2) the usual western and eastern edges to the Block, which extend beyond the swath of the instrument, (3) all actually observed areas for which there is no valid atmospheric aerosol characterization, as well as (4) the missing pixel values due to the computer issue mentioned earlier in the paper. In this particular case, and again with the data available at the time that map was generated, large land surface areas could not be processed because the aerosol product (generated at the spatial resolution of 17.6 km) was not available. This was due to the presence of clouds

315  in the area, as well as the decision to only provide an aerosol product whenever the result was of sufficient quality. Again, a better product is now available at a finer spatial resolution (4.4 km), but that is beside the point of this paper, which focuses on missing data.

- p. 8, l.2 of the caption of Figure 5: *But why is the cloud cover at 17.6km? I thought that RCCM and ASCM was at 275m?*

    Response: The image in Figure 5 is a downstream product derived from the MISR-HR sharpened L1B2 data, to which

320    atmospheric correction has been applied as part of the processing. The atmospheric correction depends on the MISR standard Level 2 aerosol product, which until very recently was produced on a 17.6 km grid. This map was derived using the MISR L2 atmospheric product "available at the time that paper [Mahlangu et al. (2018)] was generated", as indicated in the manuscript.

    Regarding the standard MISR cloud masks, please note that

325    ○ The Radiometric Camera-by-Camera Cloud Mask (RCCM) is exclusively available at the spatial resolution of 1.1 km. See the ATBD JPL D-13397, Rev. B, entitled 'Level 1 Cloud Detection Algorithm Theoretical Basis', by David J. Diner, Larry Di Girolamo, and Eugene E. Clothiaux, available from https://eospso.gsfc.nasa.gov/sites/default/files/atbd/atbd-misr-06.pdf.

    ○ The Angular Signature Cloud Mask (ASCM) product is also generated at the spatial resolution of 1.1 km. See the

330    ATBD JPL D-81127, Revision A, entitled 'Data Products Specifications for the MISR Level 2 Classifiers Product', by Catherine Moroney, Larry DiGirolamo, and Alexandra Jones, available from https://eosweb.larc.nasa.gov/sites/default/files/project/misr/DPS_Classifiers.pdf.

    Hence there has never been a MISR cloud product at the spatial resolution of 275 m.

    In summary, the presence of clouds, as documented by the 1.1 km cloud product, inside the 17.6 km blocks, prevented

335    the derivation of the aerosol product in those coarser blocks, which in turn controlled the feasibility of deriving surface products.

- p. 9, l. 171: *I thought that NASA removed this after the end of February 2020?*

    Response: It is not clear whether this question relates to the data policy or the FTP access, so we will address both aspects:

340    ○ There has not been any change in the NASA data policy, and there are no plans to do so in the foreseeable future. What does happen from time to time is that the Distributed Active Archive Centers (DAACs) update or modernize

their web portals: the current implementation is called "EarthData" and is accessible at https://l0dup05.larc.nasa. gov/MISR/cgi-bin/MISR/main.cgi, in the case of ASDC. Users need to register once to get access to that system, and thereafter can browse, order and receive data without even needing a password. This setup aims in part to collect metrics about how much data is ordered and where it goes, but there are no data access restrictions.

○ All NASA DAACs, and ASDC in particular, are contemplating moving away from FTP and providing access to data through HTTPS instead, probably at some point in 2020. This is a purely technical change, and ample documentation (with examples) will be provided in due time to help users. In any case, FTP is still available as of this writing, though the Data Pool is accessible only after registering into the EarthData portal.

To ensure that the paper remains current in the future, we have updated the text to read 'As is the case for other Earth Observation missions, all MISR data products are freely available to anyone interested, either through customized orders, or through bulk downloads from the Data Pool at https://eosweb.larc.nasa.gov/project/misr/misr_table.'

- p. 10, l. 197: *and over clouds*

Response: As already mentioned above, atmospheric products may use either the ellipsoid or the terrain-projected radiance product, depending on geographical location. Note also that the expression 'over oceans' does not restrictively refers to oceanic applications, but to all applications (including the atmosphere) where the ocean constitutes the lower boundary. To be consistent with this viewpoint, the manuscript has been modified to read: 'reference ellipsoid, primarily used in applications over oceans, and one where measurements are projected on a Digital Elevation Model (DEM), primarily used over land.' (rather than 'in terrestrial applications').

- p. 10, l. 214: *other and delete 'others'*

Response: The text has been changed to read "among the other 35 simultaneously acquired,".

- p. 11, line 236: *to*

Response: The word 'to' has been changed to 'into'.

- p. 12, Figure 6: These would be much more useful if they employed a colour-density lookup table to show the relative importance of the off-achromatic pixels.

Response: We do not understand this comment because it is not clear what "off-achromatic pixels" are, especially because each scatterplot involves only one spectral band, so there is no "chromaticity" involved. In any case, those scatterplots are merely provided to give a visual impression of the general consistency between the measurements obtained in the indicated data channels. The quantitative evidence supporting the use of the proposed algorithm lies in the statistical characteristics of those relations (correlation coefficients, linear fits, RMSD and numerical estimates of dispersion), not in qualitative images.

- p. 14, l. 255: *How is this software operational? Does it run at the DAAC?*

Response: The manuscript describes an algorithm that replaces missing values in MISR L1B2 GRP data files as explained, and documents a set of computer codes that implement this process. These software routines are 'operational' in the sense that the input data are available from NASA and that the source code is publicly available from GitHub, but users must process their own L1B2 data using those tools. This correcting system has not (yet) been implemented at NASA ASDC. See our answer to Reviewer #1 for more details on this issue. To avoid any possible confusion, this and the next sentences have been modified and merged to read "which implements these concepts in the open source IDL processing software Version 2.2.0 available from GitHub."

- p. 15, line 258: *Insert a comma*

  Response: A comma has been inserted, as suggested.

- p. 15, line 259: *Replace 'Brf by BRF'*

  Response: This suggestion is understandable but has not been implemented because we wanted to specifically distinguish between the ToA Brf product generated by MISR (discussed in this paper) from the BoA BRF product generated by the MISR-HR processing system (documented in other existing and upcoming publications). In any case, the nomenclature 'Brf' has been used consistently throughout the paper, having been defined in Equation 1.

- p. 16, l. 310: *It would be helpful to add some performance figures here. Also, will this be taken up by NASA in the planned level-1 re-processing?*

  Response: (A) In the vast majority of cases, 1 or 2 iterations will replace the bulk of the missing values, unless the best source data channel happens to also have missing data for the same geographical locations. It is only in this latter case that the number of "attempts" is increased, and even then, 4 is sufficient. The case presented in Subsection 4.3 (BA camera for P168, O002111 and B110) is really exceptional: we have not encountered any other similar (or worse) situation yet...

  (B) The issue of having NASA ASDC implementing this algorithm has been discussed and is envisaged for the next funding phase, but has not been formally approved yet. See the discussion of this point above.

- p. 19, l. 329: *than*

  Response: The word 'than' has been inserted into the manuscript, as suggested.

- p. 24, l. 350: *Why not use MODIS BRFs from MOD09 or Landsat-7 taken at the same time as MISR??*

  Response: Comparisons of measurements acquired by different instruments are fraught with difficulties because of significant differences in spectral band responses, point spread functions, sensitivity to perturbing factors (such as stray light, or angular effects), and the timing of observations. Such comparisons amount to a cross-calibration study, which is interesting in its own right, but fall outside the scope of this paper.

  The original MODIS instrument is also hosted on the same Terra platform, but uses a very different sensor technology, specifically a multispectral cross-track scanning radiometer. Because it relies on a rotating mirror to gather reflected light

405  from different directions onto the detectors, pixel size varies drastically with scan angle (bow tie effect) and this has to be corrected after the fact. Direct comparison with MISR's pushbroom technology is difficult until one works with higher level products. In any case, the MODIS MOD09 data product reports on the surface reflectance (Bottom of Atmosphere, BoA) while the MISR L1B2 GRP data product discussed in this paper is a Top of Atmosphere (ToA) product, generated before any atmospheric correction is applied. Those products cannot be meaningfully compared.

410  As for Landsat, most readily available products are also surface products, and the spatial resolution is much finer (30 m instead of 275 or 1100 m). Simple averaging of Landsat pixels is possible but would not necessarily be comparable to MISR's measurements without extensive additional processing.

The approach proposed here, which consists of artificially declaring some of the available data missing, is the most practical and definitive because the values estimated by the algorithm should match as closely as possible the original
415  values which are known: there is no need to look for another instrument to provide reference values, since they are available in the data.

- p. 26, l. 363: *a*

Response: The word 'a' has been deleted, as suggested.

---

## Author Response (AR1)

24 March 2020

Dear ESSD Editors,

The final version of the manuscript essd-2019-210 has been uploaded on the web site under the name 'MISR_L1B2_v8.pdf'. This version incorporates all appropriate suggestions made by the anonymous Reviewers #1 and #2. The PDF file attached to the "Authors' Comments" (https://doi.org/10.5194/essd-2019-210-AC1) contains our point-by-point responses to all remarks received.

Please note that
* All maps, graphs and tables were generated by myself using the public domain software described in this paper.
* All MISR input data are publicly available at NASA, as indicated in the paper.

I have never had much success with the software tool LaTeXDiff: it systematically generates error messages on my computer. However, I found a similar (free) service on the web, actually provided by a German company (https://tools.pdf24.org/en/compare-pdf), which allows comparing two PDF documents on-line. I have saved the output as a MS Word document, and subsequently reformatted it in PDF. The resulting document is provided below: it is not as readable as a regular paper, but all changes between the initially submitted manuscript and the most recent final version are highlighted in color, so it is actually quite simple to witness what has changed.

I hope these will satisfy your expectations. I am looking forward to hear from you. Best personal regards, Michel Verstraete.

[revised manuscript text omitted]

---

## Author Response (AR2)

30 April 2020

Dear Kirsten and ESSD Colleagues,

I have now uploaded the LaTeX file for the manuscript essd-2019-210, as well as the figures and the BiBTeX file. Please note the comments to the typesetter inserted near the start of the LaTeX source file, and maintain the specific choices of font types, as they carry particular meanings within the context of this paper.

I am still waiting for the release of the latest version of the MISR Toolkit for Windows computers: my NASA JPL colleague had promised it for the end of last week, so I assume this will be imminent. As soon as we get this essential library, we will generate the Virtual Machine code to be made available through GFZ (on web page 10.5880/fidgeo.2020.011), together with its dedicated user manual.

Best regards, Michel Verstraete.